# Graph-Based Operator Learning from Limited Data on Irregular Domains

## Abstract

Operator learning seeks to approximate mappings from input functions to output solutions, particularly in the context of partial differential equations (PDEs). While recent advances such as DeepONet and Fourier Neural Operator (FNO) have demonstrated strong performance, they often rely on regular grid discretizations, limiting their applicability to complex or irregular domains. In this work, we propose a **G**raph-based **O**perator **L**earning with **A**ttention (GOLA) framework that addresses this limitation by constructing graphs from irregularly sampled spatial points and leveraging attention-enhanced Graph Neural Netwoks (GNNs) to model spatial dependencies with global information. To improve the expressive capacity, we introduce a Fourier-based encoder that projects input functions into a frequency space using learnable complex coefficients, allowing for flexible embeddings even with sparse or nonuniform samples. We evaluated our approach across a range of 2D PDEs, including Darcy Flow, Advection, Eikonal, and Nonlinear Diffusion, under varying sampling densities. Our method consistently outperforms baselines, particularly in data-scarce regimes, demonstrating strong generalization and efficiency on irregular domains.

## 1 Introduction

Learning mappings between function spaces is a fundamental task in computational physics and scientific machine learning, especially for approximating solution operators of partial differential equations (PDEs). Operator learning offers a paradigm shift by learning the solution operator directly from data, enabling fast, mesh-free predictions across varying input conditions. Despite their success, existing operator learning models such as DeepONet (Lu et al., 2019) and Fourier Neural Operator (FNO) (Li et al., 2020a) exhibit notable limitations that restrict their applicability in more general settings. A key shortcoming lies in their reliance on regular, uniform grid discretizations. FNO, for instance, requires inputs to be defined on fixed Cartesian grids to leverage fast Fourier transforms efficiently. This assumption limits their flexibility and generalization ability when applied to problems defined on complex geometries, irregular meshes, or unstructured domains, which are common in real-world physical systems. Furthermore, these models often struggle with sparse or non-uniformly sampled data, leading to degraded performance and increased computational cost when adapting to more realistic, heterogeneous scenarios.

To address these limitations, we propose a **G**raph-based **O**perator **L**earning with **A**ttention (GOLA) framework that leverages Graph Neural Networks (GNNs) to learn PDE solution operators over irregular spatial domains. By constructing graphs from sampled spatial coordinates and encoding local geometric and functional dependencies through message passing, the model naturally adapts to non-Euclidean geometries. To enhance global expressivity, we further incorporate attention-based mechanisms that can capture long-range dependencies more effectively and a Fourier-based encoder that projects input functions into a frequency domain using learnable complex-valued bases. Our model exhibits superior data efficiency and generalization, achieving smaller prediction errors with fewer training samples and demonstrating robustness under domain shifts.

The main contributions of this work are as follows:

- We introduce GOLA, a unified architecture combining spectral encoding and attention-enhanced GNNs for operator learning on irregular domains.

- We propose a learnable Fourier encoder that projects input functions into a frequency domain tailored for spatial graphs.

- Through extensive experiments, we demonstrate that GOLA generalizes across PDE types, sample densities, and resolution shifts, achieving state-of-the-art performance in challenging data-scarce regimes.

## 2 Related Work

There are many latest research about graph and attention methods in scientific machine learning (Xiao et al., 2024), (Kissas et al., 2022), (Boullé and Townsend, 2024), (Xu et al., 2024), (Jin and Gu, 2023), (Cuomo et al., 2022) (Kovachki et al., 2024), (Nelsen and Stuart, 2024), (Batlle et al., 2023). Despite these advances, most existing approaches either assume structured discretizations or focus primarily on either graph-based locality or attention-based global modeling, but rarely integrate both in a unified operator-learning framework designed specifically for irregularly sampled domains. In particular, there remains a lack of architectures that jointly leverage spectral representations and attention-enhanced graph reasoning for data-efficient operator learning under sparse and non-uniform sampling.

**Graph neural networks for scientific machine learning.** (Battaglia et al., 2018) applies shared functions over nodes and edges, captures relational inductive biases and generalizes across different physical scenarios. (Bar-Sinai et al., 2019) learns data-driven discretization schemes for solving PDEs by training a neural network to predict spatial derivatives directly from local stencils. By replacing hand-crafted finite difference rules with learned operators, it adapts discretizations to the underlying data for improved accuracy and generalization. (Sanchez-Gonzalez et al., 2020) predicts future physical states by performing message passing over the mesh graph, capturing both local and global dynamics without relying on explicit numerical solvers. Graph Kernel Networks (GKNs) (Li et al., 2020b) directly approximates continuous mappings between infinite-dimensional function spaces by utilizing graph kernel convolution layers. PDE-GCN (Wang et al., 2022) represents partial differential equations on arbitrary graphs by combining spectral graph convolution with PDE-specific inductive biases. It learns to predict physical dynamics directly on graph-structured domains, enabling generalization across varying geometries and discretizations. The Message Passing Neural PDE Solver (Brandstetter et al., 2022) formulates spatiotemporal PDE dynamics by applying learned message passing updates on graph representations of the solution domain. Physics-Informed Transformer (PIT) (Dos Santos et al., 2023) embeds physical priors into the Transformer architecture to model PDE surrogate solutions. It leverages self-attention to capture long-range dependencies and integrates PDE residuals as soft constraints during training to improve generalization. GraphCast (Lam et al., 2024) learns the Earth's atmosphere as a spatiotemporal graph and uses a graph neural network to iteratively forecast future weather states based on past observations. It performs message passing over the graph to capture spatial correlations and temporal dynamics, enabling accurate medium-range forecasts.

However, while these graph-based methods effectively model relational inductive biases and local geometric structures, they often lack explicit spectral representations that enable global functional approximation across irregular domains.

**Attention-based methods for scientific machine learning.** U-Netformer (Liu et al., 2022) proposes a hybrid neural architecture that combines the U-Net's hierarchical encoder-decoder structure with transformer-based attention modules to capture both local and global dependencies in PDE solution spaces. Tokenformer (Zhou et al., 2023) reformulates PDE solving as a token mixing problem by representing input fields as tokens and applying self-attention across them to model spatial correlations. Adaptive Fourier Neural Operators (AFNO) (Guibas et al., 2021) are an efficient token-mixing mechanism for vision transformers that perform resolution-independent global convolution in the Fourier domain which is enhanced by block-diagonal channel mixing, adaptive weight sharing, and frequency sparsification to deliver quasi-linear complexity and superior performance over traditional self-attention on high-resolution image tasks. There is still a gap in designing operator-learning architectures that simultaneously maintain resolution indepen-

dence, spectral expressivity, and graph-based adaptability while remaining data-efficient in sparse-sample regimes.

The proposed GOLA architecture integrates three complementary components: (i) a learnable Fourier encoder that injects global spectral structure into the representation, (ii) graph-based message passing that captures local geometric relationships on irregular domains, and (iii) multi-head self-attention and attention-weighted aggregation that enable global dependency modeling across nodes. This unified design allows the model to simultaneously exploit spectral priors, relational inductive biases, and long-range interactions while remaining applicable to irregularly sampled spatial domains.

# 3 Methodology

## 3.1 Problem Formulation

Consider the general form of a PDE

$$\mathcal{N}[u](x) = f(x), \quad x \in \Omega \tag{1}$$

where $x$ denotes spatial coordinates, $\Omega$ is the spatial domain. $\mathcal{N}$ is a differential operator, $u(x)$ is the unknown solution, and $f(x)$ is a given source term.

The objective is to learn the solution operator $\mathcal{G} : \mathcal{F} \to \mathcal{U}$, where $\mathcal{F}$ and $\mathcal{U}$ are Banach spaces of input functions and solution functions, respectively. For each PDE instance, we assume $f \in \mathcal{F}, \quad u \in \mathcal{U}$ such that $u = \mathcal{G}(f)$. We assume access to a training dataset $\mathcal{D} = \{(f_n, u_n)\}_{n=1}^{N}$, consisting of multiple input-output function pairs, where each $f_n(\cdot)$ and $u_n(\cdot)$ is represented by discrete samples over a finite set of points.

While existing approaches such as FNO have demonstrated strong performance, they typically rely on structured, grid-based discretizations of the domain. This assumption limits their applicability to unstructured meshes, complex geometries, and adaptively sampled domains. To overcome this limitation, we employ GNNs for operator learning by representing the domain as a graph. This allows for modeling on arbitrary domains and sampling patterns. Once trained, the operator learning model can efficiently predict the solution $u$ for a new instance of the input $f$ at random locations.

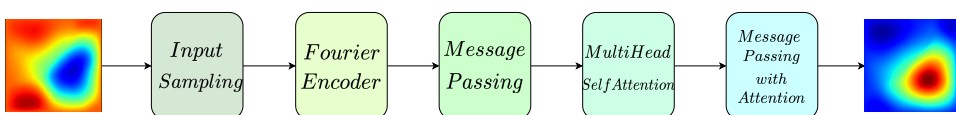

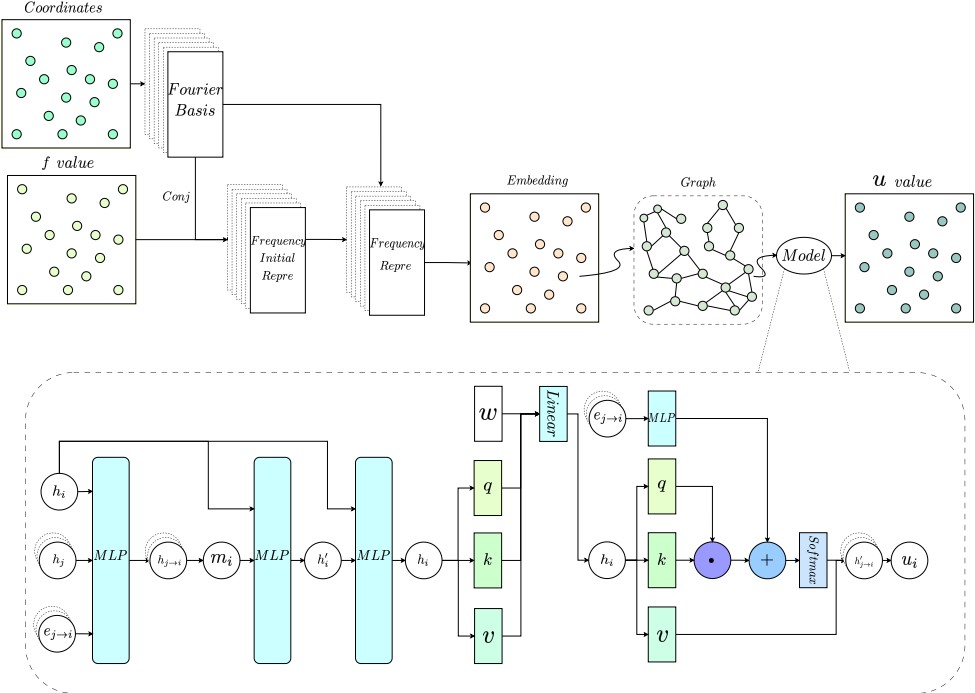

Figure 1: *GOLA: Graph-based Operator Learning with Attention.* The model first encodes input function values sampled on irregular spatial coordinates using a learnable Fourier encoder to obtain spectral node features. A graph is constructed based on spatial proximity, enabling message passing and multi-head self-attention to capture local and global dependencies. A final attention-based message passing layer refines the representation to predict the output solution values. We combine global basis projection, local interaction and global correction. GOLA effectively handles irregular domains and sparse samples, achieving strong generalization for PDE operator learning.

## 3.2 Graph Construction

Existing neural operators such as FNO rely on structured grids, which limits their applicability to irregular domains. The reasons why we constrcut graph are as follows: (i) Irregular spatial samples naturally form a point cloud; (ii) Graphs provide permutation invariance and geometric flexibility; (iii) Message passing enables local relational inductive bias; (iv) Graphs allow resolution-agnostic generalization. Thus, the graph formulation directly addresses the limitation of grid-dependent neural operators.

To represent PDE solutions over irregular domains, we assume that each input function is observed at a finite set of spatial locations $\{x_i\}_{i=1}^N \subset \Omega$. These points arise from sampling processes, such as measurement sensors, adaptive meshes, or unstructured discretizations. We represent the sampled domain as a graph $G = (V, E)$, where each node $v_i \in V$ corresponds to a spatial coordinate $x_i$, and edges encode spatial relationships between nearby points. Edges are created based on spatial proximity. Two nodes are connected if the Euclidean distance between them is less than a threshold $r$ such that $(i, j) \in E$ if and only if $\|x_i - x_j\|_2 \leq r$. Each edge $(i, j)$ carries edge attributes $e_{ij}$ that encode both geometric and feature-based information, such as the relative coordinates and function values at nodes $i$ and $j$ such that $e_{ij} = Concat\,(x_i, x_j, f(x_i), f(x_j))$, This graph-based representation allows us to model unstructured spatial domains and enables message passing among nonuniform samples.

## 3.3 Fourier Encoder

Classical FNO performs global convolution in Fourier space but requires uniform grids for FFT. The motivations why we design Fourier Encoder are that (i) Fourier bases provide a global functional prior; (ii) Spectral representations are resolution-independent; (iii) Learnable frequencies allow adaptation to irregular sampling; (iv) Fourier Encoder injects global structure before local message passing. Without this spectral

encoder, purely local GNNs struggle to capture long-range dependencies efficiently, especially in low-data regimes. Thus, the Fourier Encoder introduces global inductive bias while remaining mesh-free.

We define a set of learnable frequencies $\{\omega_m \in \mathbb{R}^2 \mid m = 1, \ldots, M\}$.

For any coordinate $x \in \mathbb{R}^2$, the $m$-th basis function is given by the complex exponential

$$\varphi_m(x) = e^{2\pi i \langle \omega_m, x \rangle} \tag{2}$$

where $\langle \cdot, \cdot \rangle$ denotes the standard Euclidean inner product, and $i$ is the imaginary unit.

At the discrete level, for a batch of $B$ samples and $N$ points per sample, the basis matrix is defined as

$$\Phi \in \mathbb{C}^{B \times N \times M}, \quad \Phi_{b,i,m} = e^{2\pi i \langle \omega_m, x_i^{(b)} \rangle} \tag{3}$$

where $x_i^{(b)}$ denotes the $i$-th coordinate point in the $b$-th batch sample.

Let $f \in \mathcal{F}$ denote the input function defined over the spatial domain $\Omega$. In practice, the function is observed at a finite set of spatial coordinates $\{x_i\}_{i=1}^N \subset \Omega$. The sampled function values are represented as

$$f_N = \{f(x_i)\}_{i=1}^N. \tag{4}$$

For a batch of $B$ samples with $C_{\text{in}}$ input feature channels per spatial point, the discretized observations are organized into a tensor

$$f \in \mathbb{R}^{B \times C_{\text{in}} \times N}, \tag{5}$$

where $f_{b,c,i}$ denotes the value of the $c$-th input channel at spatial location $x_i$ for the $b$-th sample.

Given the input $f \in \mathbb{R}^{B \times C_{\text{in}} \times N}$ sampled at points $\{x_i\}$, we first project onto the Fourier basis. We compute the Fourier coefficients by

$$\hat{u}_{b,c,m} = \frac{1}{N} \sum_{i=1}^N f_{b,c,i} \, \overline{\varphi_m\left(x_i^{(b)}\right)} \tag{6}$$

where $\overline{(\cdot)}$ denotes complex conjugation.

We introduce a learnable set of complex Fourier coefficients $W \in \mathbb{C}^{C_{\text{in}} \times C_{\text{out}} \times M}$, where $C_{out}$ is the number of output feature channels produced by the Fourier encoder. The spectral filtering operation is

$$\hat{v}_{b,o,m} = \sum_{c=1}^{C_{\text{in}}} \hat{u}_{b,c,m} \, W_{c,o,m} \tag{7}$$

We reconstruct the output in the physical domain by applying the inverse transform

$$v_{b,o,i} = \sum_{m=1}^M \hat{v}_{b,o,m} \, \varphi_m\left(x_i^{(b)}\right) \tag{8}$$

Since $v$ is complex-valued, we only take its real part for the output as

$$h = \text{Re}(v) \in \mathbb{R}^{B \times C_{\text{out}} \times N}. \tag{9}$$

The output $h$ serves as the input node features for the downstream GNN model.

### 3.4 Message Passing

Standard GNNs are limited by local receptive fields, over-smoothing with depth, and difficulty modeling global interactions. Message passing mechanisms are particularly effective for modeling strong local geometric structures.

Given a node $i \in V$ and its set of neighbors $\mathcal{N}(i)$, the pre-processed messages $\{m_{ij}\}_{j \in \mathcal{N}(i)}$ are first computed using a learnable neural network $g_\Theta$ as

$$m_{ij} = g_\Theta(h_i, h_j, e_{ij}) \tag{10}$$

where $h_i$ and $h_j$ are node features, and $e_{ij}$ denotes edge attributes.

Then we aggregate message from neighbors (Corso et al., 2020) such that

$$\mu_i = \frac{1}{|\mathcal{N}(i)|} \sum_{j \in \mathcal{N}(i)} m_{ij}. \tag{11}$$

$$\hat{m}_i = \text{Concat}\left(\mu_i, \; \max_{j \in \mathcal{N}(i)} m_{ij}, \; \min_{j \in \mathcal{N}(i)} m_{ij}, \; \sqrt{\frac{1}{|\mathcal{N}(i)|} \sum_{j \in \mathcal{N}(i)} (m_{ij} - \mu_i)^2}\right). \tag{12}$$

This concatenated feature vector is processed by a post-aggregation neural network $\gamma_\Theta$ to produce the updated node representation by

$$h_i' = \gamma_\Theta\left(h_i, \hat{m}_i\right) \tag{13}$$

The updated node representation is passed through additional MLP layers with residual connections to enhance expressiveness.

### 3.5 Multi-Head Self-Attention

Multi-head self-attention enables the modeling of global dependencies by allowing each node to attend to all others. We employ $L$ independent attention heads. For each head $l$, the query, key and value functions are computed as linear projections

$$q^{(l)}(z) = W_q h'(z), \quad k^{(l)}(y) = W_k h'(y), \quad v^{(l)}(y) = W_v h'(y) \tag{14}$$

where $y, z \in \mathbb{R}^2$ denote spatial coordinates, $d_l$ is the dimension per attention head, $W_q, W_k, W_v \in \mathbb{R}^{d_l \times C_{\text{out}}}$, $q^{(l)}(z), k^{(l)}(y), v^{(l)}(y) \in \mathbb{R}^{d_l}$ are learned head-specific features.

Before computing attention, the keys and values are normalized

$$\tilde{k}^{(l)}(y) = \text{Norm}(k^{(l)}(y)), \quad \tilde{v}^{(l)}(y) = \text{Norm}(v^{(l)}(y)) \tag{15}$$

where $\text{Norm}(\cdot)$ denotes instance normalization. We compute connection between key and value by

$$G_l = \sum_{j=1}^{N} \tilde{k}^{(l)}(y_j)^\top \tilde{v}^{(l)}(y_j) w, \tag{16}$$

where $G_l \in \mathbb{R}^{d_l \times d_l}$, $w$ is the weight calculated by the number of points $N$.

We define $\mathcal{K}_l$ for each head as a function as follows:

$$\mathcal{K}_l(z) = q^{(l)}(z) G_l \tag{17}$$

The outputs are concatenated and projected to the output space by

$$\mathcal{K}(z) = \text{Concat}\left(\mathcal{K}_1(z), \ldots, \mathcal{K}_L(z)\right) \tag{18}$$

$$\hat{h}(z) = W_{\text{out}}\mathcal{K}(z), \tag{19}$$

where $W_{\text{out}} \in \mathbb{R}^{C_{\text{out}} \times (C_{\text{out}} \cdot L)}$.

The result is then passed through a linear projection layer to update the node features.

### 3.6 Message Passing with Attention

Attention-weighted aggregation introduces adaptivity by assigning different importance weights to neighboring nodes during feature aggregation.

The attention weights $\alpha_{ij}$ between node $i$ and node $j$ are computed using a scaled dot-product attention mechanism by

$$\alpha_{ij} = \text{softmax}_{j \in \mathcal{N}(i)}\left(\frac{\left(W_4\hat{h}_i\right)^\top \left(W_5\hat{h}_j + W_3 e_{ij}\right)}{\sqrt{L}}\right). \tag{20}$$

Then we update node features and add a skip connection by

$$\hat{h}'_i = W_1\hat{h}_i + \sum_{j \in \mathcal{N}(i)} \alpha_{ij}\left(W_2\hat{h}_j + W_3 e_{ij}\right) + W_s\hat{h}_i \tag{21}$$

Then we add a linear projection to produce the predicted solution $\hat{u}$.

### 3.7 Training

The model is trained to minimize the relative $L_2$ error between predicted and true solutions by

$$Loss = \frac{\|u - \mathcal{G}(f)\|_{L^2(\Omega)}}{\|u\|_{L^2(\Omega)}} \tag{22}$$

Under sparse sampling and limited training data, purely local models struggle to extrapolate global structure. The spectral encoder supplies a compact global representation, while attention enhances expressivity without requiring deep stacking. Empirically, this design leads to improved generalization and lower error in data-scarce regimes.

## 4 Theoretical Analysis

Following the universal approximation theorem for operators (Lu et al., 2019), neural operator architectures can approximate any continuous operator $\mathcal{N}$ between Banach spaces when provided with sufficient capacity.

**Proposition.** As notations in Section 3.1, let $\mathcal{N} : \mathcal{F} \to \mathcal{U}$ be a continuous nonlinear operator between separable Banach spaces. We assume the solution operator maps into $\mathcal{U} \subset L^2(\Omega)$ which allows the framework to accommodate functions that may exhibit discontinuities while remaining square-integrable. Let $\mathcal{F}_\delta$ denote a compact subset of the input function space $\mathcal{F}$. Then, under sufficient model capacity, the GOLA architecture $\mathcal{G}$ can approximate $\mathcal{N}$ well in the $L^2(\Omega)$ norm over a compact domain $\Omega$, i.e., $\sup_{f \in \mathcal{F}_\delta} \|\mathcal{N}(f) - \mathcal{G}(f)\|_{L^2(\Omega)} < \epsilon$, for any $\epsilon > 0$ and compact subset $\mathcal{F}_\delta \subset \mathcal{F}$.

*Proof.* Given a function $f \in \mathcal{F} \subset L^2(\Omega)$, we sample it at $N$ spatial locations $\{x_i\}_{i=1}^N \subset \Omega$ to obtain a discrete representation $f_N = (f(x_1), \ldots, f(x_N)) \in \mathbb{R}^N$. We assume these points are sampled from a distribution

whose support is the domain $\Omega$. Under this assumption, as $N \to \infty$, the empirical point set becomes dense in $\Omega$ with high probability. Therefore, the discrete representation $f_N = (f(x_1), \ldots, f(x_N))$ provides an increasingly accurate approximation of $f$ in $L^2(\Omega)$. Thus, $f_N$ can approximate $f$ well in $L^2(\Omega)$ norm via interpolation over the sampling set.

As defined in Eq. 2, there is a set of complex Fourier basis functions $\{\varphi_m(x) = e^{2\pi i \langle \omega_m, x \rangle}\}_{m=1}^{M}$. The Fourier basis is complete in $L^2(\Omega)$, so for any $f \in \mathcal{F}$ and $\delta > 0$, there exists $M$ such that

$$\left\| f(x) - \sum_{m=1}^{M} \hat{u}_m \varphi_m(x) \right\|_{L^2(\Omega)} < \delta.$$

where $\hat{u}_m$ is defined at Eq. 6. This guarantees that the learnable Fourier encoder in GOLA can approximate the functional input $f$ to any precision.

As defined in Section 3.2, we construct a graph $G = (V, E)$ with node set $V = \{x_i\}_{i=1}^{N}$, where edges encode local spatial relationships. According to universal approximation results for GNNs (Xu et al., 2019), (Morris et al., 2019), for any continuous function defined on graphs, a GNN with sufficient depth and width can approximate it well. Thus, the GNN decoder can approximate the mapping from input features to solution values

$$(f(x_1), \ldots, f(x_N)) \mapsto (\mathcal{N}(f)(x_1), \ldots, \mathcal{N}(f)(x_N))$$

Let $\mathcal{T}_N$ denote the sampling operator, $\mathcal{F}_\theta$ the Fourier encoder, and $\mathcal{D}_\theta$ the GNN decoder. Then the GOLA operator can be written as

$$\mathcal{G}_\theta = \mathcal{D}_\theta \circ \mathcal{F}_\theta \circ \mathcal{T}_N$$

Each component is continuous and approximates its target arbitrarily well. Since composition of continuous approximations preserves continuity, and $\mathcal{F}_\delta$ is compact, the total approximation error can be made less than any $\varepsilon > 0$ by choosing $N$, $M$, and model capacity large enough such that

$$\sup_{f \in \mathcal{F}_\delta} \|\mathcal{N}(f) - \mathcal{G}(f)\|_{L^2(\Omega)} < \varepsilon$$

## 5 Experiments

We evaluate the proposed model GOLA on four 2D PDE benchmarks including Darcy Flow, Nonlinear Diffusion, Eikonal, and Advection. For each dataset, we simulate training data with 5, 10, 20, 30, 40, 50, 80, 100 samples and use 100 examples for testing. To construct graphs, we randomly sample 20, 30, 40, 50, 60, 70, 80, 90, 100, 200, 300, 400, 500, 600, 700, 800, 900, 1000 points from a uniform $128 \times 128$ grid over the domain $[0, 1] \times [0, 1]$. The sampled points define the nodes of the graph. Our model learns to approximate the solution operator from these irregularly sampled inputs. We aim to test generalization under both limited data and resolution changes. We compare against the following baselines including DeepONet (Lu et al., 2019), AFNO (Guibas et al., 2021) and Graph Kernel Network (GKN) (Li et al., 2020b).

**Comparisons with baselines.** Table 1 reports the averaged test errors over 5 runs with different seeds across four PDE benchmarks—Darcy Flow, Advection, Eikonal, and Nonlinear Diffusion—in the low-data regime of 100 training samples with sample density = 1000 randomly selected from a uniform $128 \times 128$ grid over the domain $[0, 1] \times [0, 1]$. The proposed GOLA method consistently achieves the lowest error across all datasets. For Darcy Flow, GOLA attains an error of $0.1088 \pm 0.0027$, representing a 40.8% relative improvement over the best baseline, GKN ($0.1840 \pm 0.0040$). In Advection, GOLA achieves $0.2227 \pm 0.0185$, reducing the error by 26.7% compared to GKN and by over 77% relative to AFNO and DeepONet. For Eikonal, GOLA obtains $0.0657 \pm 0.0011$, a 45.7% improvement over GKN, while Nonlinear Diffusion exhibits the largest relative gain—$0.0430 \pm 0.0005$, which is 59.2% lower than GKN. Moreover, GOLA maintains standard deviations on par with or below those of the best-performing baselines, indicating both superior accuracy and stable convergence.

Table 1: *Test errors for different models in irregular sampling points trained on 100 training data samples with sample density=1000 across various PDE benchmarks. The results are averaged over 5 runs in this paper.*

| Dataset | AFNO | DeepONet | GKN | Ours(GOLA) |
|---|---|---|---|---|
| Darcy Flow | $0.4310 \pm 0.0040$ | $0.5897 \pm 0.0026$ | $0.1840 \pm 0.0040$ | $\mathbf{0.1088 \pm 0.0027}$ |
| Advection | $0.9845 \pm 0.0007$ | $0.9979 \pm 0.0001$ | $0.3043 \pm 0.0041$ | $\mathbf{0.2227 \pm 0.0185}$ |
| Eikonal | $0.1828 \pm 0.0017$ | $0.1918 \pm 0.0004$ | $0.1210 \pm 0.0043$ | $\mathbf{0.0657 \pm 0.0011}$ |
| Nonlinear Diffusion | $0.1686 \pm 0.0016$ | $0.2781 \pm 0.0005$ | $0.1052 \pm 0.0038$ | $\mathbf{0.0430 \pm 0.0005}$ |

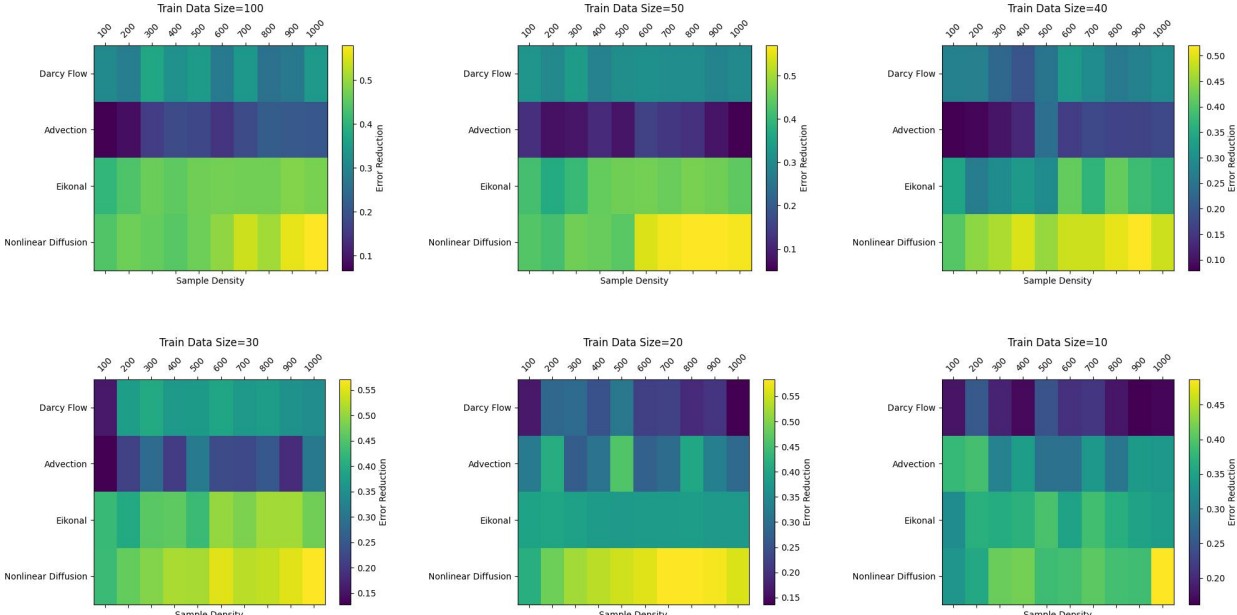

Figure 2: *Error reduction heatmaps across training data sizes and sample densities for PDE Benchmarks. Nonlinear Diffusion consistently shows the highest error reduction across all training sizes and densities and it becomes more prominent at high sample densities even under very small training size 10.*

**Generalization across sample densities.** From Table 2, we use 100 training data, and choose three types of sampling densities 20, 500, 1000 which represent small, medium and high sample densities. We observe a consistent trend that increasing sample density leads to significant performance improvements across all PDEs. The results highlight that higher sampling density substantially improves generalization, particularly for PDEs with more complex solution manifolds such as Darcy flow and nonlinear diffusion, and that even moderate densities 500 are sufficient to close much of the performance gap for Eikonal equations.

Table 2: *Test errors for small, medium, and high sampling densities with training data size=100.*

| Sample Density | 20 | 500 | 1000 |
|---|---|---|---|
| Darcy flow | $0.4422 \pm 0.0213$ | $0.1298 \pm 0.0043$ | $0.1088 \pm 0.0027$ |
| Advection | $0.4374 \pm 0.0177$ | $0.2654 \pm 0.0163$ | $0.2227 \pm 0.0185$ |
| Eikonal | $0.1267 \pm 0.0019$ | $0.0675 \pm 0.0020$ | $0.0657 \pm 0.0011$ |
| Nonlinear diffusion | $0.1901 \pm 0.0060$ | $0.0542 \pm 0.0015$ | $0.0430 \pm 0.0005$ |

**Resolution generalization.** From Table 3 and Figure 3, we use 100 training data and sample 1000 training sample points, then we test the relative $L_2$ error in different test sample densities 100, 500, 1000, 2000, 4000. We observe that higher test sample densities consistently reduce the error for all PDE families, reflecting improved approximation accuracy with denser test points.

Table 3: *Test errors for different test sampling densities with training sample density=1000.*

| Test Sample Density | 100 | 500 | 1000 | 2000 | 4000 |
|---|---|---|---|---|---|
| Darcy flow | $0.2475 \pm 0.0041$ | $0.1304 \pm 0.0020$ | $0.1088 \pm 0.0027$ | $0.0971 \pm 0.0033$ | $0.0895 \pm 0.0035$ |
| Advection | $0.3641 \pm 0.0117$ | $0.2505 \pm 0.0149$ | $0.2227 \pm 0.0185$ | $0.2218 \pm 0.0202$ | $0.2182 \pm 0.0141$ |
| Eikonal | $0.0790 \pm 0.0031$ | $0.0672 \pm 0.0020$ | $0.0657 \pm 0.0011$ | $0.0654 \pm 0.0024$ | $0.0654 \pm 0.0019$ |
| Nonlinear diffusion | $0.0893 \pm 0.0020$ | $0.0511 \pm 0.0015$ | $0.0430 \pm 0.0005$ | $0.0386 \pm 0.0012$ | $0.0368 \pm 0.0015$ |

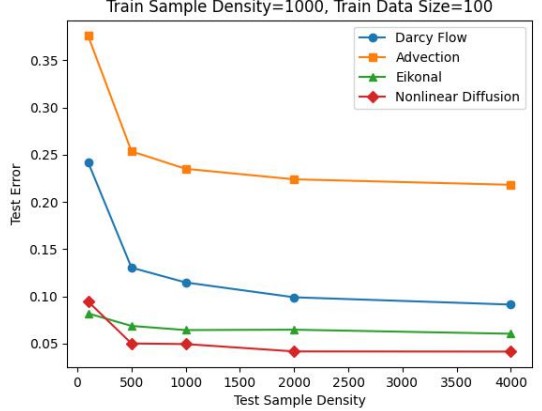

Figure 3: *Test error trend with test sample density*

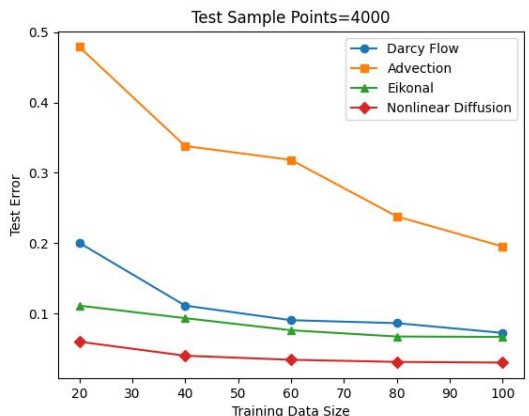

Figure 4: *Test error trend with train data size*

**Data Efficiency.** From Table 4, we use 2000 sample points and change different training data size to test the performance. From Figure 4, we report the results for 4000 sample points with different training data size. In Figure 5, we report the results for test error trend with respect to training data size in test sample points $\in \{200, 300, 400, 500, 600, 700, 800, 900\}$. Across all PDEs, we observe a clear trend of decreasing test error with increasing training data size, indicating effective data scaling behavior.

Table 4: *Test errors under varying numbers of training data size with sample density=2000.*

| Training data size | 20 | 40 | 60 | 80 | 100 |
|---|---|---|---|---|---|
| Darcy flow | $0.2027 \pm 0.0161$ | $0.1372 \pm 0.0095$ | $0.1071 \pm 0.0073$ | $0.0983 \pm 0.0057$ | $0.0913 \pm 0.0029$ |
| Advection | $0.5253 \pm 0.0273$ | $0.4026 \pm 0.0182$ | $0.3192 \pm 0.0388$ | $0.2709 \pm 0.0243$ | $0.2228 \pm 0.0172$ |
| Eikonal | $0.1029 \pm 0.0047$ | $0.0763 \pm 0.0033$ | $0.0678 \pm 0.0028$ | $0.0648 \pm 0.0023$ | $0.0647 \pm 0.0021$ |
| Nonlinear diffusion | $0.0815 \pm 0.0139$ | $0.0538 \pm 0.0023$ | $0.0429 \pm 0.0036$ | $0.0394 \pm 0.0033$ | $0.0360 \pm 0.0013$ |

**Graph Visualizations.** We visualize graph construction in Figure 6. We randomly sample 1000 node positions in the unit square and use ball connectivity with a fixed radius 0.2 to construct graph. These results are shown on the top row. Then in this graph, we visualize input function values on the graph, ground-truth solution, and model prediction on the bottom row. Figure 6 demonstrates that (i) the graph construction preserves locality and global connectivity; (ii) the learned model generalizes well to unseen node configurations and accurately reconstructs the solution field; (iii) visual comparison between ground truth and predictions reveals minimal discrepancy, supporting the effectiveness of our proposed model GOLA.

**Time Complexity and Memory Cost.** We analyze the computational complexity of the GOLA architecture in terms of the number of spatial points $N$, Fourier modes $M$, dimension of node features in the graph $C$, and edges $E \sim \mathcal{O}(Nk)$, where $k$ is the average number of neighbors in the sparse spatial graph. The time complexity for GOLA is $\mathcal{O}(MNC) + \mathcal{O}(NkC^2) + \mathcal{O}(NkC)$. The count of parameters for GOLA is 2,900,249.

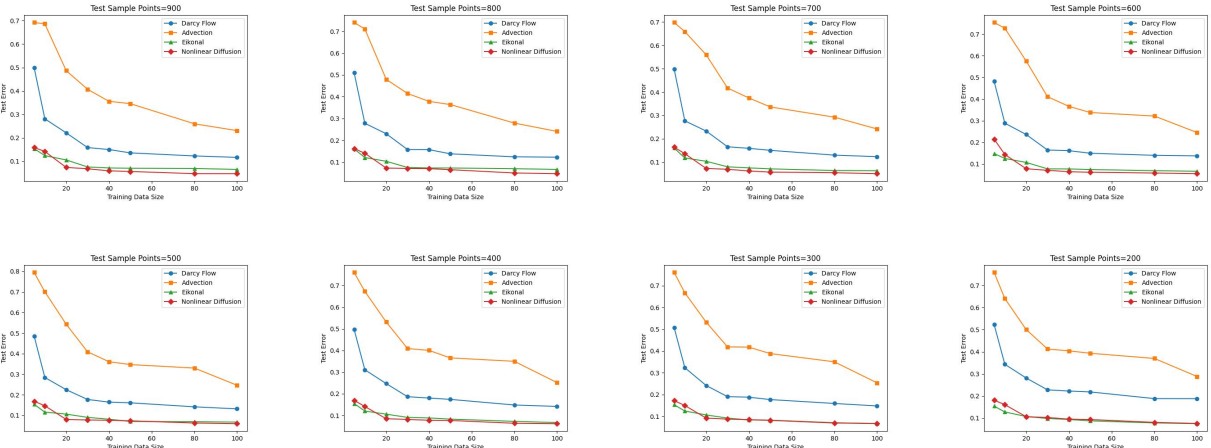

Figure 5: *Test error trends across varying sample densities for PDE benchmarks.*

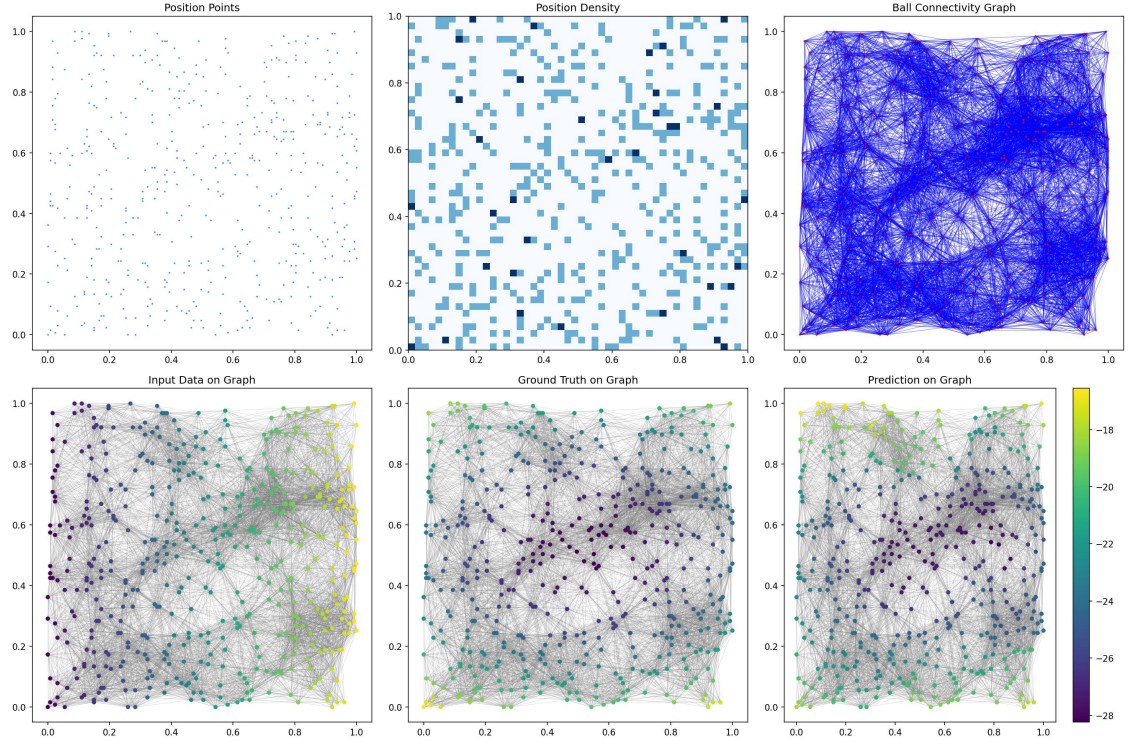

Figure 6: *Visualizations for graph with 1000 sample points on Advection.*

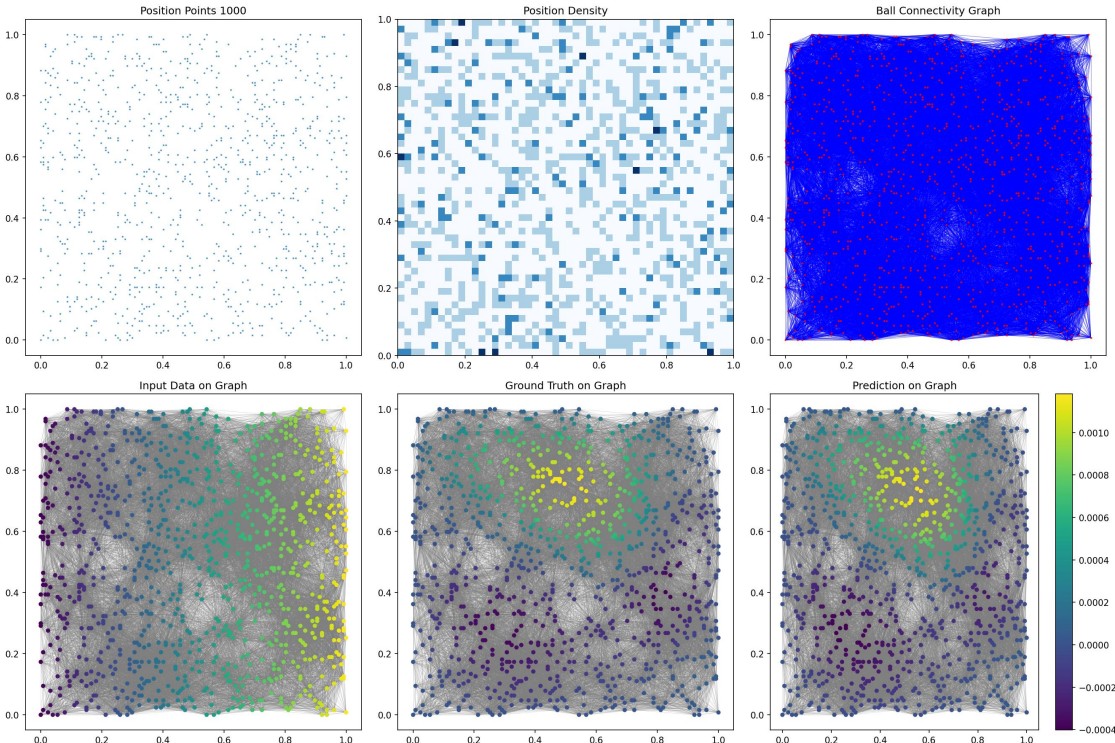

Figure 7: Graph Visulizations for Darcy.

# 6 Conclusion

In this work, We introduce **G**raph-based **O**perator **L**earning with **A**ttention (GOLA) framework, which combines a learnable Fourier encoder with attention-enhanced message passing to solve PDEs over irregular domains. By representing the spatial domain as a proximity graph and embedding inputs into a learnable spectral basis, GOLA effectively captures both local and global dependencies, enabling accurate operator approximation even under sparse sampling and complex geometries. Through comprehensive experiments across diverse PDE benchmarks including Darcy Flow, Advection, Eikonal, and Nonlinear Diffusion, GOLA consistently outperforms baselines including AFNO, DeepONet, GKN particularly in data-scarce regimes. We demonstrate GOLA's superior generalization, resolution scalability, and robustness to sparse sampling. These results highlight the potential of combining spectral encoding and localized message passing with attention to build continuous, data-efficient operator approximators that adapt naturally to non-Euclidean geometries. This study demonstrates that graph-based representations provide a powerful and flexible foundation for advancing operator learning in real-world physical systems with irregular data.

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

# A  Summary of Notations in GOLA

Table 5: Summary of Notations in GOLA

| | | |
|---|---|---|
| **PDE Formulation** | | |
| $\Omega$ | Domain | Spatial domain of the PDE |
| $x$ | $\mathbb{R}^2$ | Spatial coordinates |
| $\mathcal{N}[\cdot]$ | Operator | Differential operator in the PDE |
| $f(\cdot)$ | Function | Input function |
| $u(\cdot)$ | Function | PDE solution |
| $\mathcal{G} : \mathcal{F} \to \mathcal{U}$ | Neural operator | Learned operator |
| $N$ | Integer | Number of points per sample |
| $\mathcal{D} = \{(f^n, u^n)\}_{n=1}^N$ | Dataset | Training dataset |
| **Graph Construction** | | |
| $V = \{x_i\}_{i=1}^N$ | Node set | Sampled spatial points |
| $e_{ij}$ | Vector | Edge feature between node $i$ and $j$ |
| $E$ | Edge set | Edges defined by radius threshold |
| $G = (V, E)$ | Graph | Spatial graph representation |
| $r$ | Scalar | Radius for connectivity |
| $\mathcal{N}(i)$ | Set | Neighborhood of node $i$ |
| **Fourier Encoder** | | |
| $M$ | Integer | Number of Fourier modes |
| $\omega_m$ | $\mathbb{R}^2$ | Learnable frequency vector |
| $\varphi_m(\cdot)$ | Function | Fourier basis $e^{2\pi i \langle \omega_m, x \rangle}$ |
| $B$ | Integer | Batch size |
| $\Phi$ | $\mathbb{C}^{B \times N \times M}$ | Fourier basis matrix |
| $\hat{u}$ | $\mathbb{C}^{B \times C_{in} \times M}$ | Fourier coefficients of input |
| $\hat{v}$ | $\mathbb{C}^{B \times C_{out} \times M}$ | Coefficients for inverse transform |
| $W$ | $\mathbb{C}^{C_{in} \times C_{out} \times M}$ | Learnable spectral weights |
| $h$ | $\mathbb{R}^{B \times C_{out} \times N}$ | Node feature after inverse transform |
| **Message Passing** | | |
| $h_i$ | Vector | Node feature |
| $m_{ij}$ | Vector | Message from node $j$ to node $i$ |
| $g_\Theta$ | Neural Network | Message function |
| $\hat{m}_i$ | Vector | Aggregated neighbor message |
| $\gamma_\Theta$ | Neural Network | Update function |
| $h_i'$ | Vector | Updated node representation for node $i$ |
| **Multi-Head Self-Attention** | | |
| $h'(z)$ | Vector | Updated node feature for the point at $z$ |
| $L$ | Integer | Number of attention heads |
| $d_l$ | Integer | Feature dimension per head |
| $W_q, W_k, W_v$ | Matrices | Query, key, value projections |
| $q^{(l)}(z), k^{(l)}(y), v^{(l)}(y)$ | Vectors | Head-specific projections at a specific point |
| $G_l$ | Matrix | Matrix describing key and value |
| $\mathcal{K}_l(\cdot)$ | Function | Query representation with key and value |
| $\hat{h}(z)$ | Vector | Updated node feature for the point at $z$ |

| Message Passing with Attention | | |
|---|---|---|
| $W_1, W_2, W_3, W_4, W_5, W_s$ | Matrices | Learnable parameters |
| $\hat{h}_i$ | Vector | Input feature of node $i$ |
| $\hat{h}_i'$ | Vector | Updated node feature after attention aggregation for node $i$ |
| $\alpha_{ij}$ | Scalar | Attention weight assigned to neighbor $j$ for node $i$ |
| $\hat{u}(\cdot)$ | Function | Predicted PDE solution |
| Analysis and Complexity | | |
| $C$ | Interger | Dimension of node features in the graph |
| $k$ | Integer | Average neighbors |

# B    Additional Results

## B.1    Test error trends across varying sample densities(30-100) for PDE benchmarks

In Figure 8, it shows the test error for four PDE benchmarks as a function of training data size, under different test point resolutions ranging from 30 to 100.

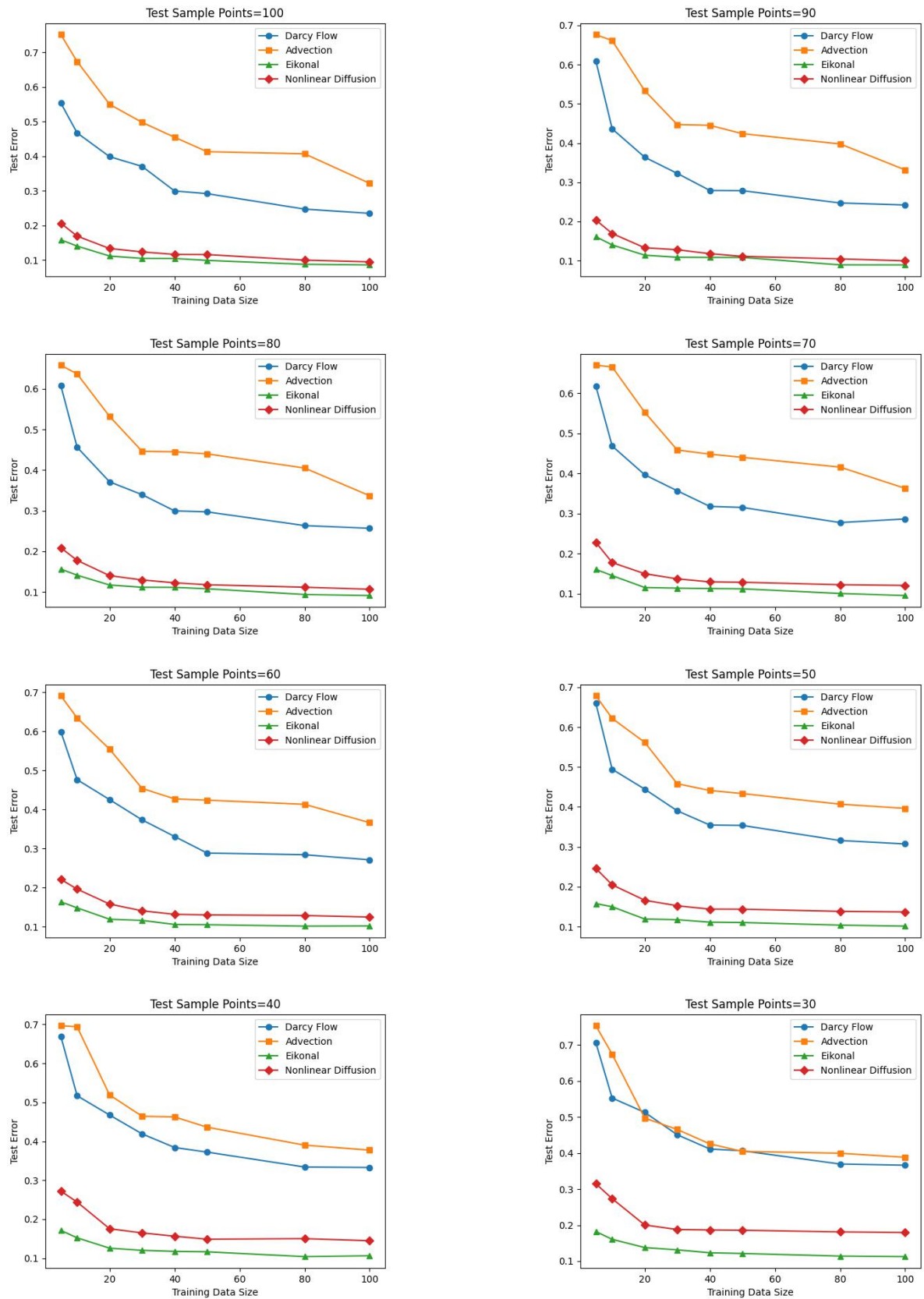

Figure 8: Test error trends across varying sample densities(30-100) for PDE benchmarks

## B.2   Error reduction on 30, 50, 100 training data across various sampling density

From Table 6, 7, 8, with 100, 50, 30 training data size respectively, for each PDE benchmark, we choose sample density from 20 to 1000 to compare GKN and GOLA, and calculate the error reduction. It shows that our method is better than GKN and error reduction is significant.

Table 6: Test errors trained on 100 training data size across various sampling density

| (a) *Darcy Flow* | | | |
|---|---|---|---|
| Density | GKN | Ours | Error Reduction |
| 20 | 0.5027 | 0.4073 | 18.98% |
| 30 | 0.4746 | 0.3663 | 22.82% |
| 40 | 0.4572 | 0.3328 | 27.21% |
| 50 | 0.4283 | 0.3072 | 28.27% |
| 60 | 0.3995 | 0.2711 | 32.14% |
| 70 | 0.3885 | 0.2710 | 30.24% |
| 80 | 0.3760 | 0.2566 | 31.76% |
| 90 | 0.3646 | 0.2420 | 33.63% |
| 100 | 0.3402 | 0.2349 | 30.95% |
| 200 | 0.2633 | 0.1878 | 28.67% |
| 300 | 0.2355 | 0.1479 | 37.20% |
| 400 | 0.2097 | 0.1415 | 32.52% |
| 500 | 0.2020 | 0.1315 | 34.90% |
| 600 | 0.1911 | 0.1242 | 35.01% |
| 700 | 0.1874 | 0.1235 | 34.10% |
| 800 | 0.1788 | 0.1226 | 31.43% |
| 900 | 0.1777 | 0.1168 | 34.27% |
| 1000 | 0.1748 | 0.1147 | 34.38% |

| (b) *Eikonal* | | | |
|---|---|---|---|
| Density | GKN | Ours | Error Reduction |
| 20 | 0.1808 | 0.1236 | 31.64% |
| 30 | 0.1723 | 0.1125 | 34.71% |
| 40 | 0.1671 | 0.1066 | 39.80% |
| 50 | 0.1620 | 0.0981 | 39.44% |
| 60 | 0.1588 | 0.0962 | 39.42% |
| 70 | 0.1546 | 0.0953 | 38.36% |
| 80 | 0.1472 | 0.0911 | 38.11% |
| 90 | 0.1442 | 0.0871 | 39.47% |
| 100 | 0.1442 | 0.0857 | 40.57% |
| 200 | 0.1325 | 0.0745 | 43.77% |
| 300 | 0.1238 | 0.0664 | 46.37% |
| 400 | 0.1203 | 0.0658 | 45.30% |
| 500 | 0.1205 | 0.0640 | 46.89% |
| 600 | 0.1199 | 0.0632 | 47.29% |
| 700 | 0.1159 | 0.0614 | 47.02% |
| 800 | 0.1148 | 0.0607 | 47.13% |
| 900 | 0.1173 | 0.0606 | 48.34% |
| 1000 | 0.1168 | 0.0611 | 47.69% |

| (c) *Nonlinear Diffusion* | | | |
|---|---|---|---|
| Density | GKN | Ours | Error Reduction |
| 20 | 0.2407 | 0.1958 | 18.65% |
| 30 | 0.2181 | 0.1795 | 17.70% |
| 40 | 0.2133 | 0.1446 | 32.21% |
| 50 | 0.2066 | 0.1369 | 33.74% |
| 60 | 0.1922 | 0.1248 | 35.07% |
| 70 | 0.1876 | 0.1206 | 35.71% |
| 80 | 0.1851 | 0.1067 | 42.36% |
| 90 | 0.1760 | 0.0995 | 43.47% |
| 100 | 0.1689 | 0.0947 | 43.93% |
| 200 | 0.1415 | 0.0755 | 46.64% |
| 300 | 0.1245 | 0.0674 | 45.86% |
| 400 | 0.1118 | 0.0618 | 44.72% |
| 500 | 0.1115 | 0.0594 | 46.73% |
| 600 | 0.1093 | 0.0553 | 49.41% |
| 700 | 0.1101 | 0.0507 | 53.95% |
| 800 | 0.1073 | 0.0469 | 56.29% |
| 900 | 0.1054 | 0.0463 | 56.07% |
| 1000 | 0.1044 | 0.0439 | 57.95% |

| (d) *Advection* | | | |
|---|---|---|---|
| Density | GKN | Ours | Error Reduction |
| 20 | 0.9035 | 0.3980 | 55.95% |
| 30 | 0.5409 | 0.3883 | 28.21% |
| 40 | 0.4563 | 0.3773 | 17.31% |
| 50 | 0.4208 | 0.3747 | 10.96% |
| 60 | 0.3823 | 0.3662 | 4.21% |
| 70 | 0.3736 | 0.3626 | 2.94% |
| 80 | 0.3468 | 0.3362 | 2.88% |
| 90 | 0.3422 | 0.3315 | 3.13% |
| 100 | 0.3446 | 0.3216 | 6.67% |
| 200 | 0.3150 | 0.2880 | 8.57% |
| 300 | 0.3017 | 0.2537 | 15.91% |
| 400 | 0.3084 | 0.2516 | 18.42% |
| 500 | 0.2997 | 0.2462 | 17.85% |
| 600 | 0.2886 | 0.2453 | 14.47% |
| 700 | 0.2972 | 0.2421 | 18.54% |
| 800 | 0.3038 | 0.2408 | 21.89% |
| 900 | 0.2926 | 0.2310 | 21.05% |
| 1000 | 0.2886 | 0.2290 | 20.65% |

Table 7: Test errors trained on 50 training data size across various sampling densities

**(a) *Darcy Flow***

| Density | GKN | Ours | Error Reduction |
|---|---|---|---|
| 20 | 0.5690 | 0.4484 | 21.20% |
| 30 | 0.5192 | 0.4064 | 21.73% |
| 40 | 0.5085 | 0.3722 | 26.80% |
| 50 | 0.4783 | 0.3536 | 26.07% |
| 60 | 0.4467 | 0.2884 | 35.44% |
| 70 | 0.4328 | 0.3099 | 28.40% |
| 80 | 0.4310 | 0.2934 | 31.93% |
| 90 | 0.4099 | 0.2785 | 32.06% |
| 100 | 0.4068 | 0.2756 | 32.25% |
| 200 | 0.3092 | 0.2182 | 29.43% |
| 300 | 0.2673 | 0.1771 | 33.74% |
| 400 | 0.2421 | 0.1741 | 28.09% |
| 500 | 0.2326 | 0.1609 | 30.83% |
| 600 | 0.2173 | 0.1497 | 31.11% |
| 700 | 0.2114 | 0.1470 | 30.46% |
| 800 | 0.1990 | 0.1381 | 30.60% |
| 900 | 0.1901 | 0.1360 | 28.46% |
| 1000 | 0.1890 | 0.1328 | 29.74% |

**(b) *Eikonal***

| Density | GKN | Ours | Error Reduction |
|---|---|---|---|
| 20 | 0.1908 | 0.1361 | 28.67% |
| 30 | 0.1870 | 0.1213 | 35.13% |
| 40 | 0.1839 | 0.1162 | 36.81% |
| 50 | 0.1764 | 0.1103 | 37.47% |
| 60 | 0.1739 | 0.1052 | 39.51% |
| 70 | 0.1670 | 0.1024 | 38.68% |
| 80 | 0.1640 | 0.1000 | 39.02% |
| 90 | 0.1643 | 0.0943 | 42.60% |
| 100 | 0.1592 | 0.0927 | 41.77% |
| 200 | 0.1386 | 0.0873 | 37.01% |
| 300 | 0.1380 | 0.0824 | 40.29% |
| 400 | 0.1341 | 0.0741 | 44.74% |
| 500 | 0.1292 | 0.0702 | 45.67% |
| 600 | 0.1309 | 0.0705 | 46.14% |
| 700 | 0.1299 | 0.0713 | 45.11% |
| 800 | 0.1288 | 0.0688 | 46.58% |
| 900 | 0.1300 | 0.0704 | 45.85% |
| 1000 | 0.1273 | 0.0710 | 44.23% |

**(c) *Nonlinear Diffusion***

| Density | GKN | Ours | Error Reduction |
|---|---|---|---|
| 20 | 0.2817 | 0.2202 | 21.83% |
| 30 | 0.2554 | 0.1858 | 27.25% |
| 40 | 0.2413 | 0.1485 | 38.46% |
| 50 | 0.2339 | 0.1437 | 38.56% |
| 60 | 0.2266 | 0.1303 | 42.50% |
| 70 | 0.2117 | 0.1285 | 39.30% |
| 80 | 0.2081 | 0.1177 | 43.44% |
| 90 | 0.1955 | 0.1110 | 43.22% |
| 100 | 0.1903 | 0.1083 | 43.09% |
| 200 | 0.1597 | 0.0932 | 41.64% |
| 300 | 0.1506 | 0.0818 | 45.68% |
| 400 | 0.1390 | 0.0766 | 44.89% |
| 500 | 0.1301 | 0.0733 | 43.66% |
| 600 | 0.1349 | 0.0618 | 54.19% |
| 700 | 0.1308 | 0.0575 | 56.04% |
| 800 | 0.1284 | 0.0552 | 57.01% |
| 900 | 0.1258 | 0.0542 | 56.92% |
| 1000 | 0.1218 | 0.0532 | 56.32% |

**(d) *Advection***

| Density | GKN | Ours | Error Reduction |
|---|---|---|---|
| 20 | 0.8899 | 0.4338 | 51.25% |
| 30 | 0.6991 | 0.4046 | 42.13% |
| 40 | 0.5983 | 0.3987 | 33.36% |
| 50 | 0.5569 | 0.4052 | 27.24% |
| 60 | 0.6016 | 0.4143 | 31.13% |
| 70 | 0.4733 | 0.4067 | 14.07% |
| 80 | 0.4585 | 0.3984 | 13.11% |
| 90 | 0.4559 | 0.3819 | 16.23% |
| 100 | 0.4467 | 0.3925 | 12.13% |
| 200 | 0.4156 | 0.3845 | 7.48% |
| 300 | 0.4132 | 0.3796 | 8.13% |
| 400 | 0.4126 | 0.3658 | 11.34% |
| 500 | 0.3767 | 0.3468 | 7.94% |
| 600 | 0.3981 | 0.3378 | 15.15% |
| 700 | 0.3893 | 0.3364 | 13.59% |
| 800 | 0.3778 | 0.3304 | 12.55% |
| 900 | 0.3619 | 0.3328 | 8.04% |
| 1000 | 0.3717 | 0.3528 | 5.08% |

Table 8: Test errors trained on 30 training data size across various sampling densities

| **(a)** *Darcy Flow* | | | | | **(b)** *Eikonal* | | | |
|---|---|---|---|---|---|---|---|---|
| Density | GKN | Ours | Error Reduction | | Density | GKN | Ours | Error Reduction |
| 20 | 0.5574 | 0.4984 | 10.58% | | 20 | 0.1929 | 0.1402 | 27.32% |
| 30 | 0.5203 | 0.4509 | 13.34% | | 30 | 0.1920 | 0.1313 | 31.61% |
| 40 | 0.5028 | 0.4190 | 16.67% | | 40 | 0.1893 | 0.1200 | 36.61% |
| 50 | 0.4881 | 0.3902 | 20.06% | | 50 | 0.1852 | 0.1174 | 36.61% |
| 60 | 0.4667 | 0.3733 | 20.01% | | 60 | 0.1801 | 0.1162 | 35.48% |
| 70 | 0.4576 | 0.3565 | 22.09% | | 70 | 0.1757 | 0.1138 | 35.23% |
| 80 | 0.4568 | 0.3394 | 25.70% | | 80 | 0.1803 | 0.1115 | 38.16% |
| 90 | 0.4513 | 0.3225 | 28.54% | | 90 | 0.1817 | 0.1088 | 40.12% |
| 100 | 0.4404 | 0.3134 | 28.84% | | 100 | 0.1825 | 0.1047 | 42.63% |
| 200 | 0.3651 | 0.2275 | 37.69% | | 200 | 0.1633 | 0.0981 | 39.93% |
| 300 | 0.3168 | 0.1906 | 39.84% | | 300 | 0.1701 | 0.0921 | 45.86% |
| 400 | 0.2938 | 0.1864 | 36.56% | | 400 | 0.1684 | 0.0908 | 46.08% |
| 500 | 0.2810 | 0.1772 | 36.94% | | 500 | 0.1582 | 0.0901 | 43.05% |
| 600 | 0.2702 | 0.1646 | 39.08% | | 600 | 0.1553 | 0.0776 | 50.03% |
| 700 | 0.2618 | 0.1659 | 36.63% | | 700 | 0.1550 | 0.0764 | 50.71% |
| 800 | 0.2524 | 0.1576 | 37.56% | | 800 | 0.1541 | 0.0756 | 50.94% |
| 900 | 0.2454 | 0.1572 | 35.94% | | 900 | 0.1550 | 0.0736 | 52.52% |
| 1000 | 0.2375 | 0.1554 | 34.57% | | 1000 | 0.1530 | 0.0762 | 50.20% |
| **(c)** *Nonlinear Diffusion* | | | | | **(d)** *Advection* | | | |
| Density | GKN | Ours | Error Reduction | | Density | GKN | Ours | Error Reduction |
| 20 | 0.2958 | 0.2365 | 20.05% | | 20 | 0.9881 | 0.4798 | 51.44% |
| 30 | 0.2761 | 0.1878 | 31.98% | | 30 | 0.7924 | 0.4655 | 41.25% |
| 40 | 0.2581 | 0.1647 | 36.19% | | 40 | 0.7468 | 0.4642 | 37.81% |
| 50 | 0.2512 | 0.1523 | 39.37% | | 50 | 0.6657 | 0.4580 | 31.20% |
| 60 | 0.2423 | 0.1408 | 41.89% | | 60 | 0.6622 | 0.4537 | 31.49% |
| 70 | 0.2353 | 0.1369 | 41.82% | | 70 | 0.5756 | 0.4464 | 22.45% |
| 80 | 0.2219 | 0.1295 | 41.64% | | 80 | 0.5646 | 0.4460 | 21.01% |
| 90 | 0.2135 | 0.1279 | 40.09% | | 90 | 0.5889 | 0.4470 | 24.10% |
| 100 | 0.2168 | 0.1234 | 43.08% | | 100 | 0.5708 | 0.4457 | 21.92% |
| 200 | 0.1919 | 0.1024 | 46.64% | | 200 | 0.5242 | 0.4123 | 21.35% |
| 300 | 0.1709 | 0.0873 | 48.92% | | 300 | 0.5841 | 0.4191 | 28.25% |
| 400 | 0.1683 | 0.0815 | 51.57% | | 400 | 0.5157 | 0.4090 | 20.69% |
| 500 | 0.1588 | 0.0773 | 51.32% | | 500 | 0.5915 | 0.4089 | 30.87% |
| 600 | 0.1582 | 0.0709 | 55.18% | | 600 | 0.5344 | 0.4019 | 24.79% |
| 700 | 0.1547 | 0.0685 | 55.72% | | 700 | 0.5387 | 0.4172 | 22.55% |
| 800 | 0.1495 | 0.0703 | 52.98% | | 800 | 0.5527 | 0.4148 | 24.95% |
| 900 | 0.1519 | 0.0683 | 55.04% | | 900 | 0.4994 | 0.4074 | 18.42% |
| 1000 | 0.1520 | 0.0652 | 57.11% | | 1000 | 0.5353 | 0.3707 | 30.75% |

### B.3 Test errors across training data size for different sample density

We choose sample density from 20 to 900, and for training data size from 5 to 100, we test on four PDE benchmarks as follows.

Table 9: *Test sample density=900*

| Training data size | 5 | 10 | 20 | 30 | 40 | 50 | 80 | 100 |
|---|---|---|---|---|---|---|---|---|
| Darcy Flow | 0.4996 | 0.2812 | 0.2215 | 0.1587 | 0.1503 | 0.1360 | 0.1230 | 0.1168 |
| Advection | 0.6921 | 0.6862 | 0.4866 | 0.4074 | 0.3560 | 0.3460 | 0.2601 | 0.2310 |
| Eikonal | 0.1540 | 0.1245 | 0.1059 | 0.0760 | 0.0718 | 0.0704 | 0.0694 | 0.0655 |
| Nonlinear Diffusion | 0.1589 | 0.1420 | 0.0740 | 0.0683 | 0.0595 | 0.0567 | 0.0474 | 0.0463 |

Table 10: *Test sample density=800*

| Training data size | 5 | 10 | 20 | 30 | 40 | 50 | 80 | 100 |
|---|---|---|---|---|---|---|---|---|
| Darcy Flow | 0.5102 | 0.2781 | 0.2303 | 0.1576 | 0.1572 | 0.1381 | 0.1243 | 0.1226 |
| Advection | 0.7407 | 0.7109 | 0.4799 | 0.4148 | 0.3789 | 0.3641 | 0.2791 | 0.2408 |
| Eikonal | 0.1606 | 0.1210 | 0.1038 | 0.0756 | 0.0731 | 0.0730 | 0.0697 | 0.0625 |
| Nonlinear Diffusion | 0.1623 | 0.1406 | 0.0728 | 0.0706 | 0.0701 | 0.0652 | 0.0501 | 0.0469 |

Table 11: *Test sample density=700*

| Training data size | 5 | 10 | 20 | 30 | 40 | 50 | 80 | 100 |
|---|---|---|---|---|---|---|---|---|
| Darcy Flow | 0.4985 | 0.2768 | 0.2332 | 0.1659 | 0.1592 | 0.1506 | 0.1301 | 0.1235 |
| Advection | 0.6978 | 0.6590 | 0.5594 | 0.4172 | 0.3747 | 0.3364 | 0.2926 | 0.2421 |
| Eikonal | 0.1605 | 0.1180 | 0.1038 | 0.0801 | 0.0755 | 0.0713 | 0.0643 | 0.0635 |
| Nonlinear Diffusion | 0.1642 | 0.1363 | 0.0731 | 0.0696 | 0.0623 | 0.0575 | 0.0548 | 0.0507 |

Table 12: *Test sample density=600*

| Training data size | 5 | 10 | 20 | 30 | 40 | 50 | 80 | 100 |
|---|---|---|---|---|---|---|---|---|
| Darcy Flow | 0.4817 | 0.2886 | 0.2362 | 0.1646 | 0.1616 | 0.1497 | 0.1401 | 0.1376 |
| Advection | 0.7537 | 0.7271 | 0.5754 | 0.4109 | 0.3661 | 0.3378 | 0.3216 | 0.2453 |
| Eikonal | 0.1483 | 0.1255 | 0.1073 | 0.0776 | 0.0772 | 0.0745 | 0.0691 | 0.0665 |
| Nonlinear Diffusion | 0.2149 | 0.1440 | 0.0781 | 0.0709 | 0.0644 | 0.0618 | 0.0583 | 0.0553 |

Table 13: *Test sample density=500*

| Training data size | 5 | 10 | 20 | 30 | 40 | 50 | 80 | 100 |
|---|---|---|---|---|---|---|---|---|
| Darcy Flow | 0.4855 | 0.2842 | 0.2247 | 0.1772 | 0.1643 | 0.1609 | 0.1414 | 0.1315 |
| Advection | 0.7947 | 0.7011 | 0.5431 | 0.4089 | 0.3609 | 0.3468 | 0.3299 | 0.2462 |
| Eikonal | 0.1537 | 0.1155 | 0.1060 | 0.0901 | 0.0811 | 0.0702 | 0.0689 | 0.0671 |
| Nonlinear Diffusion | 0.1686 | 0.1464 | 0.0801 | 0.0773 | 0.0760 | 0.0733 | 0.0626 | 0.0594 |

Table 14: *Test sample density=400*

| Training data size | 5 | 10 | 20 | 30 | 40 | 50 | 80 | 100 |
|---|---|---|---|---|---|---|---|---|
| Darcy Flow | 0.4970 | 0.3109 | 0.2471 | 0.1864 | 0.1804 | 0.1741 | 0.1481 | 0.1415 |
| Advection | 0.7604 | 0.6733 | 0.5317 | 0.4090 | 0.4005 | 0.3658 | 0.3499 | 0.2516 |
| Eikonal | 0.1547 | 0.1203 | 0.1058 | 0.0908 | 0.0883 | 0.0829 | 0.0725 | 0.0669 |
| Nonlinear Diffusion | 0.1695 | 0.1411 | 0.0847 | 0.0815 | 0.0776 | 0.0766 | 0.0637 | 0.0618 |

Table 15: *Test sample density=300*

| Training data size | 5 | 10 | 20 | 30 | 40 | 50 | 80 | 100 |
|---|---|---|---|---|---|---|---|---|
| Darcy Flow | 0.5075 | 0.3233 | 0.2416 | 0.1906 | 0.1880 | 0.1771 | 0.1592 | 0.1479 |
| Advection | 0.7606 | 0.6674 | 0.5323 | 0.4191 | 0.4178 | 0.3882 | 0.3505 | 0.2537 |
| Eikonal | 0.1533 | 0.1246 | 0.1066 | 0.0921 | 0.0835 | 0.0824 | 0.0709 | 0.0664 |
| Nonlinear Diffusion | 0.1722 | 0.1498 | 0.0920 | 0.0873 | 0.0849 | 0.0818 | 0.0694 | 0.0674 |

Table 16: *Test sample density=200*

| Training data size | 5 | 10 | 20 | 30 | 40 | 50 | 80 | 100 |
|---|---|---|---|---|---|---|---|---|
| Darcy Flow | 0.5227 | 0.3431 | 0.2812 | 0.2275 | 0.2222 | 0.2182 | 0.1880 | 0.1878 |
| Advection | 0.7586 | 0.6407 | 0.5004 | 0.4123 | 0.4040 | 0.3931 | 0.3697 | 0.2880 |
| Eikonal | 0.1538 | 0.1276 | 0.1069 | 0.0981 | 0.0934 | 0.0873 | 0.0780 | 0.0745 |
| Nonlinear Diffusion | 0.1807 | 0.1605 | 0.1060 | 0.1024 | 0.0951 | 0.0932 | 0.0804 | 0.0755 |

Table 17: *Test sample density=100*

| Training data size | 5 | 10 | 20 | 30 | 40 | 50 | 80 | 100 |
|---|---|---|---|---|---|---|---|---|
| Darcy Flow | 0.5534 | 0.4673 | 0.3986 | 0.3705 | 0.2994 | 0.2918 | 0.2471 | 0.2349 |
| Advection | 0.7513 | 0.6738 | 0.5498 | 0.4977 | 0.4547 | 0.4130 | 0.4069 | 0.3216 |
| Eikonal | 0.1583 | 0.1402 | 0.1115 | 0.1047 | 0.1044 | 0.0988 | 0.0876 | 0.0857 |
| Nonlinear Diffusion | 0.2062 | 0.1691 | 0.1332 | 0.1234 | 0.1161 | 0.1157 | 0.0995 | 0.0947 |

Table 18: *Test sample density=90*

| Training data size | 5 | 10 | 20 | 30 | 40 | 50 | 80 | 100 |
|---|---|---|---|---|---|---|---|---|
| Darcy Flow | 0.6094 | 0.4362 | 0.3638 | 0.3225 | 0.2791 | 0.2785 | 0.2471 | 0.2420 |
| Advection | 0.6764 | 0.6612 | 0.5334 | 0.4470 | 0.4452 | 0.4238 | 0.3977 | 0.3315 |
| Eikonal | 0.1612 | 0.1401 | 0.1140 | 0.1088 | 0.1084 | 0.1083 | 0.0893 | 0.0891 |
| Nonlinear Diffusion | 0.2036 | 0.1693 | 0.1329 | 0.1279 | 0.1179 | 0.1110 | 0.1046 | 0.0995 |

Table 19: *Test sample density=80*

| Training data size | 5 | 10 | 20 | 30 | 40 | 50 | 80 | 100 |
|---|---|---|---|---|---|---|---|---|
| Darcy Flow | 0.6076 | 0.4561 | 0.3709 | 0.3394 | 0.2994 | 0.2970 | 0.2633 | 0.2566 |
| Advection | 0.6579 | 0.6365 | 0.5318 | 0.4460 | 0.4450 | 0.4397 | 0.4049 | 0.3368 |
| Eikonal | 0.1561 | 0.1412 | 0.1170 | 0.1115 | 0.1113 | 0.1078 | 0.0937 | 0.0911 |
| Nonlinear Diffusion | 0.2086 | 0.1777 | 0.1401 | 0.1295 | 0.1225 | 0.1177 | 0.1116 | 0.1067 |

Table 20: *Test sample density=70*

| Training data size | 5 | 10 | 20 | 30 | 40 | 50 | 80 | 100 |
|---|---|---|---|---|---|---|---|---|
| Darcy Flow | 0.6176 | 0.4682 | 0.3965 | 0.3565 | 0.3176 | 0.3151 | 0.2773 | 0.2864 |
| Advection | 0.6697 | 0.6654 | 0.5524 | 0.4583 | 0.4479 | 0.4400 | 0.4156 | 0.3626 |
| Eikonal | 0.1606 | 0.1450 | 0.1154 | 0.1138 | 0.1128 | 0.1121 | 0.1005 | 0.0953 |
| Nonlinear Diffusion | 0.2279 | 0.1776 | 0.1498 | 0.1369 | 0.1294 | 0.1285 | 0.1223 | 0.1206 |

Table 21: *Test sample density=60*

| Training data size | 5 | 10 | 20 | 30 | 40 | 50 | 80 | 100 |
|---|---|---|---|---|---|---|---|---|
| Darcy Flow | 0.5989 | 0.4762 | 0.4251 | 0.3733 | 0.3308 | 0.2884 | 0.2842 | 0.2711 |
| Advection | 0.6906 | 0.6342 | 0.5541 | 0.4537 | 0.4270 | 0.4238 | 0.4131 | 0.3662 |
| Eikonal | 0.1637 | 0.1483 | 0.1192 | 0.1162 | 0.1058 | 0.1052 | 0.1015 | 0.1020 |
| Nonlinear Diffusion | 0.2211 | 0.1963 | 0.1580 | 0.1408 | 0.1318 | 0.1303 | 0.1286 | 0.1248 |

Table 22: *Test sample density=50*

| Training data size | 5 | 10 | 20 | 30 | 40 | 50 | 80 | 100 |
|---|---|---|---|---|---|---|---|---|
| Darcy Flow | 0.6595 | 0.4944 | 0.4445 | 0.3902 | 0.3546 | 0.3536 | 0.3158 | 0.3072 |
| Advection | 0.6784 | 0.6219 | 0.5618 | 0.4580 | 0.4412 | 0.4334 | 0.4069 | 0.3747 |
| Eikonal | 0.1576 | 0.1498 | 0.1191 | 0.1174 | 0.1111 | 0.1103 | 0.1038 | 0.0981 |
| Nonlinear Diffusion | 0.2458 | 0.2048 | 0.1661 | 0.1523 | 0.1439 | 0.1437 | 0.1382 | 0.1369 |

Table 23: *Test sample density=40*

| Training data size | 5 | 10 | 20 | 30 | 40 | 50 | 80 | 100 |
|---|---|---|---|---|---|---|---|---|
| Darcy Flow | 0.6693 | 0.5169 | 0.4674 | 0.4190 | 0.3840 | 0.3722 | 0.3339 | 0.3328 |
| Advection | 0.6966 | 0.6941 | 0.5189 | 0.4642 | 0.4625 | 0.4361 | 0.3901 | 0.3773 |
| Eikonal | 0.1712 | 0.1521 | 0.1255 | 0.1200 | 0.1172 | 0.1162 | 0.1038 | 0.1060 |
| Nonlinear Diffusion | 0.2720 | 0.2439 | 0.1755 | 0.1647 | 0.1563 | 0.1485 | 0.1500 | 0.1446 |

Table 24: *Test sample density=30*

| Training data size | 5 | 10 | 20 | 30 | 40 | 50 | 80 | 100 |
|---|---|---|---|---|---|---|---|---|
| Darcy Flow | 0.7064 | 0.5529 | 0.5131 | 0.4509 | 0.4114 | 0.4064 | 0.3697 | 0.3663 |
| Advection | 0.7536 | 0.6746 | 0.4973 | 0.4655 | 0.4256 | 0.4046 | 0.3996 | 0.3883 |
| Eikonal | 0.1822 | 0.1606 | 0.1377 | 0.1313 | 0.1232 | 0.1213 | 0.1140 | 0.1125 |
| Nonlinear Diffusion | 0.3148 | 0.2735 | 0.2003 | 0.1878 | 0.1867 | 0.1858 | 0.1812 | 0.1795 |

Table 25: *Test sample density=20*

| Training data size | 5 | 10 | 20 | 30 | 40 | 50 | 80 | 100 |
|---|---|---|---|---|---|---|---|---|
| Darcy Flow | 0.7416 | 0.5963 | 0.5363 | 0.4984 | 0.4567 | 0.4484 | 0.4176 | 0.4073 |
| Advection | 0.8587 | 0.7329 | 0.5114 | 0.4798 | 0.4661 | 0.4338 | 0.4186 | 0.3980 |
| Eikonal | 0.1875 | 0.1797 | 0.1543 | 0.1402 | 0.1392 | 0.1361 | 0.1292 | 0.1236 |
| Nonlinear Diffusion | 0.3202 | 0.3074 | 0.2403 | 0.2365 | 0.2297 | 0.2202 | 0.2000 | 0.1958 |

## B.4 Error reductions for training data size=40

Table 26: *Sample density=100, train data size=40*

| Dataset | GKN | Ours | Error Reduction |
|---|---|---|---|
| Darcy Flow | 0.4122 | 0.2994 | 27.37% |
| Advection | 0.4938 | 0.4547 | 7.92% |
| Eikonal | 0.1668 | 0.1102 | 33.93% |
| Nonlinear Diffusion | 0.1947 | 0.1161 | 40.37% |
| Poisson | 0.3754 | 0.3707 | 1.25% |

Table 27: *Sample density=200, train data size=40*

| Dataset | GKN | Ours | Error Reduction |
|---|---|---|---|
| Darcy Flow | 0.3281 | 0.2388 | 27.22% |
| Advection | 0.4840 | 0.4420 | 8.68% |
| Eikonal | 0.1429 | 0.1050 | 26.52% |
| Nonlinear Diffusion | 0.1714 | 0.0951 | 44.52% |

Table 28: *Sample density=300, train data size=40*

| Dataset | GKN | Ours | Error Reduction |
|---|---|---|---|
| Darcy Flow | 0.2746 | 0.2126 | 22.58% |
| Advection | 0.4660 | 0.4178 | 10.34% |
| Eikonal | 0.1390 | 0.0982 | 29.35% |
| Nonlinear Diffusion | 0.1589 | 0.0849 | 46.57% |

Table 29: *Sample density=400, train data size=40*

| Dataset | GKN | Ours | Error Reduction |
|---|---|---|---|
| Darcy Flow | 0.2446 | 0.1968 | 19.54% |
| Advection | 0.4590 | 0.4005 | 12.75% |
| Eikonal | 0.1396 | 0.0952 | 31.81% |
| Nonlinear Diffusion | 0.1536 | 0.0776 | 49.48% |

Table 30: *Sample density=500, train data size=40*

| Dataset | GKN | Ours | Error Reduction |
|---|---|---|---|
| Darcy Flow | 0.2505 | 0.1878 | 25.03% |
| Advection | 0.4746 | 0.3609 | 23.96% |
| Eikonal | 0.1355 | 0.0957 | 29.37% |
| Nonlinear Diffusion | 0.1389 | 0.0760 | 45.28% |

Table 31: *Sample density=600, train data size=40*

| Dataset | GKN | Ours | Error Reduction |
|---|---|---|---|
| Darcy Flow | 0.2416 | 0.1648 | 31.79% |
| Advection | 0.4352 | 0.3661 | 15.88% |
| Eikonal | 0.1323 | 0.0772 | 41.65% |
| Nonlinear Diffusion | 0.1441 | 0.0739 | 48.72% |

Table 32: *Sample density=700, train data size=40*

| Dataset | GKN | Ours | Error Reduction |
|---|---|---|---|
| Darcy Flow | 0.2257 | 0.1592 | 29.46% |
| Advection | 0.4546 | 0.3747 | 17.58% |
| Eikonal | 0.1294 | 0.0819 | 36.71% |
| Nonlinear Diffusion | 0.1424 | 0.0731 | 48.67% |

Table 33: *Sample density=800, train data size=40*

| Dataset | GKN | Ours | Error Reduction |
|---|---|---|---|
| Darcy Flow | 0.2120 | 0.1572 | 25.85% |
| Advection | 0.4558 | 0.3789 | 16.87% |
| Eikonal | 0.1304 | 0.0761 | 41.64% |
| Nonlinear Diffusion | 0.1413 | 0.0701 | 50.39% |

Table 34: *Sample density=900, train data size=40*

| Dataset | GKN | Ours | Error Reduction |
|---|---|---|---|
| Darcy Flow | 0.2068 | 0.1503 | 27.32% |
| Advection | 0.4292 | 0.3560 | 17.05% |
| Eikonal | 0.1306 | 0.0805 | 38.36% |
| Nonlinear Diffusion | 0.1441 | 0.0691 | 52.05% |

Table 35: *Sample density=1000, train data size=40*

| Dataset | GKN | Ours | Error Reduction |
|---|---|---|---|
| Darcy Flow | 0.2257 | 0.1592 | 29.46% |
| Advection | 0.4546 | 0.3747 | 17.58% |
| Eikonal | 0.1294 | 0.0819 | 36.71% |
| Nonlinear Diffusion | 0.1424 | 0.0731 | 48.67% |

Table 36: *Sample density=20, train data size=40*

| Dataset | GKN | Ours | Error Reduction |
|---|---|---|---|
| Darcy Flow | 0.5527 | 0.4567 | 17.37% |
| Advection | 0.9735 | 0.5219 | 46.39% |
| Eikonal | 0.1868 | 0.1451 | 22.32% |
| Nonlinear Diffusion | 0.2892 | 0.2297 | 20.57% |

Table 37: *Sample density=30, train data size=40*

| Dataset | GKN | Ours | Error Reduction |
|---|---|---|---|
| Darcy Flow | 0.5180 | 0.4114 | 20.58% |
| Advection | 0.7502 | 0.4694 | 37.43% |
| Eikonal | 0.1847 | 0.1354 | 26.69% |
| Nonlinear Diffusion | 0.2638 | 0.1867 | 29.23% |

Table 38: *Sample density=40, train data size=40*

| Dataset | GKN | Ours | Error Reduction |
|---|---|---|---|
| Darcy Flow | 0.5074 | 0.3840 | 24.32% |
| Advection | 0.6341 | 0.4714 | 25.66% |
| Eikonal | 0.1833 | 0.1234 | 32.68% |
| Nonlinear Diffusion | 0.2464 | 0.1563 | 36.57% |

Table 39: *Sample density=50, train data size=40*

| Dataset | GKN | Ours | Error Reduction |
|---|---|---|---|
| Darcy Flow | 0.4819 | 0.3546 | 26.42% |
| Advection | 0.5321 | 0.4897 | 7.97% |
| Eikonal | 0.1798 | 0.1194 | 33.59% |
| Nonlinear Diffusion | 0.2417 | 0.1439 | 40.46% |

Table 40: *Sample density=60, train data size=40*

| Dataset | GKN | Ours | Error Reduction |
|---|---|---|---|
| Darcy Flow | 0.4664 | 0.3308 | 29.07% |
| Advection | 0.5688 | 0.4993 | 12.22% |
| Eikonal | 0.1777 | 0.1181 | 33.54% |
| Nonlinear Diffusion | 0.2349 | 0.1318 | 43.89% |

Table 41: *Sample density=70, train data size=40*

| Dataset | GKN | Ours | Error Reduction |
|---|---|---|---|
| Darcy Flow | 0.4518 | 0.3176 | 29.70% |
| Advection | 0.5410 | 0.4766 | 11.90% |
| Eikonal | 0.1663 | 0.1147 | 31.03% |
| Nonlinear Diffusion | 0.2195 | 0.1294 | 41.05% |

Table 42: *Sample density=80, train data size=40*

| Dataset | GKN | Ours | Error Reduction |
|---|---|---|---|
| Darcy Flow | 0.4558 | 0.2994 | 34.31% |
| Advection | 0.5175 | 0.4535 | 12.37% |
| Eikonal | 0.1642 | 0.1132 | 31.06% |
| Nonlinear Diffusion | 0.2127 | 0.1225 | 42.41% |

Table 43: *Sample density=90, train data size=40*

| Dataset | GKN | Ours | Error Reduction |
|---|---|---|---|
| Darcy Flow | 0.4378 | 0.2791 | 36.25% |
| Advection | 0.5039 | 0.4452 | 11.65% |
| Eikonal | 0.1706 | 0.1122 | 34.23% |
| Nonlinear Diffusion | 0.1996 | 0.1179 | 40.93% |

## B.5 Error reductions for training data size=20

Table 44: *Sample density=100, train data size=20*

| Dataset | GKN | Ours | Error Reduction |
|---|---|---|---|
| Darcy Flow | 0.4790 | 0.3986 | 16.78% |
| Advection | 0.8125 | 0.5498 | 32.33% |
| Eikonal | 0.1848 | 0.1115 | 39.66% |
| Nonlinear Diffusion | 0.2284 | 0.1332 | 41.68% |

Table 45: *Sample density=200, train data size=20*

| Dataset | GKN | Ours | Error Reduction |
|---|---|---|---|
| Darcy Flow | 0.3937 | 0.2812 | 28.58% |
| Advection | 0.8552 | 0.5004 | 41.49% |
| Eikonal | 0.1780 | 0.1069 | 39.94% |
| Nonlinear Diffusion | 0.2056 | 0.1060 | 48.44% |

Table 46: *Sample density=300, train data size=20*

| Dataset | GKN | Ours | Error Reduction |
|---|---|---|---|
| Darcy Flow | 0.3414 | 0.2416 | 29.23% |
| Advection | 0.7282 | 0.5323 | 26.90% |
| Eikonal | 0.1757 | 0.1066 | 39.33% |
| Nonlinear Diffusion | 0.1923 | 0.0920 | 52.16% |

Table 47: *Sample density=400, train data size=20*

| Dataset | GKN | Ours | Error Reduction |
|---|---|---|---|
| Darcy Flow | 0.3291 | 0.2471 | 24.92% |
| Advection | 0.7667 | 0.5317 | 30.65% |
| Eikonal | 0.1716 | 0.1058 | 38.34% |
| Nonlinear Diffusion | 0.1831 | 0.0847 | 53.74% |

Table 48: *Sample density=500, train data size=20*

| Dataset | GKN | Ours | Error Reduction |
|---|---|---|---|
| Darcy Flow | 0.3285 | 0.2247 | 31.60% |
| Advection | 1.0101 | 0.5431 | 46.23% |
| Eikonal | 0.1703 | 0.1060 | 37.76% |
| Nonlinear Diffusion | 0.1783 | 0.0801 | 55.08% |

Table 49: *Sample density=600, train data size=20*

| Dataset | GKN | Ours | Error Reduction |
|---|---|---|---|
| Darcy Flow | 0.3030 | 0.2362 | 22.05% |
| Advection | 0.7948 | 0.5754 | 27.60% |
| Eikonal | 0.1739 | 0.1073 | 38.30% |
| Nonlinear Diffusion | 0.1780 | 0.0781 | 56.12% |

Table 50: *Sample density=700, train data size=20*

| Dataset | GKN | Ours | Error Reduction |
|---|---|---|---|
| Darcy Flow | 0.3001 | 0.2332 | 22.29% |
| Advection | 0.7928 | 0.5594 | 29.44% |
| Eikonal | 0.1682 | 0.1038 | 38.29% |
| Nonlinear Diffusion | 0.1755 | 0.0731 | 58.35% |

Table 51: *Sample density=800, train data size=20*

| Dataset | GKN | Ours | Error Reduction |
|---|---|---|---|
| Darcy Flow | 0.2841 | 0.2303 | 18.94% |
| Advection | 0.8037 | 0.4799 | 40.29% |
| Eikonal | 0.1696 | 0.1038 | 38.80% |
| Nonlinear Diffusion | 0.1749 | 0.0728 | 58.38% |

Table 52: *Sample density=900, train data size=20*

| Dataset | GKN | Ours | Error Reduction |
|---|---|---|---|
| Darcy Flow | 0.2787 | 0.2215 | 20.52% |
| Advection | 0.7267 | 0.4866 | 33.04% |
| Eikonal | 0.1705 | 0.1059 | 37.89% |
| Nonlinear Diffusion | 0.1755 | 0.0740 | 57.83% |

Table 53: *Sample density=1000, train data size=20*

| Dataset | GKN | Ours | Error Reduction |
|---|---|---|---|
| Darcy Flow | 0.2660 | 0.2297 | 13.65% |
| Advection | 0.7421 | 0.5268 | 29.01% |
| Eikonal | 0.1697 | 0.1059 | 37.60% |
| Nonlinear Diffusion | 0.1716 | 0.0757 | 55.89% |

Table 54: *Sample density=20, train data size=20*

| Dataset | GKN | Ours | Error Reduction |
|---|---|---|---|
| Darcy Flow | 0.5909 | 0.5363 | 9.24% |
| Advection | 1.0704 | 0.5114 | 52.22% |
| Eikonal | 0.1985 | 0.1543 | 22.27% |
| Nonlinear Diffusion | 0.3053 | 0.2403 | 21.29% |

Table 55: *Sample density=30, train data size=20*

| Dataset | GKN | Ours | Error Reduction |
|---|---|---|---|
| Darcy Flow | 0.5712 | 0.5131 | 10.17% |
| Advection | 1.0391 | 0.4973 | 52.14% |
| Eikonal | 0.1941 | 0.1377 | 29.06% |
| Nonlinear Diffusion | 0.2804 | 0.2003 | 28.57% |

Table 56: *Sample density=40, train data size=20*

| Dataset | GKN | Ours | Error Reduction |
|---|---|---|---|
| Darcy Flow | 0.5493 | 0.4674 | 14.91% |
| Advection | 0.9612 | 0.5189 | 46.02% |
| Eikonal | 0.1916 | 0.1255 | 34.50% |
| Nonlinear Diffusion | 0.2655 | 0.1755 | 33.90% |

Table 57: *Sample density=50, train data size=20*

| Dataset | GKN | Ours | Error Reduction |
|---|---|---|---|
| Darcy Flow | 0.5221 | 0.4445 | 14.86% |
| Advection | 0.9652 | 0.5618 | 41.79% |
| Eikonal | 0.1880 | 0.1191 | 36.65% |
| Nonlinear Diffusion | 0.2525 | 0.1661 | 34.22% |

Table 58: *Sample density=60, train data size=20*

| Dataset | GKN | Ours | Error Reduction |
|---|---|---|---|
| Darcy Flow | 0.5013 | 0.4251 | 15.20% |
| Advection | 0.8556 | 0.5541 | 35.24% |
| Eikonal | 0.1864 | 0.1192 | 36.05% |
| Nonlinear Diffusion | 0.2459 | 0.1580 | 35.75% |

Table 59: *Sample density=70, train data size=20*

| Dataset | GKN | Ours | Error Reduction |
|---|---|---|---|
| Darcy Flow | 0.4975 | 0.3965 | 20.30% |
| Advection | 0.8128 | 0.5524 | 32.04% |
| Eikonal | 0.1838 | 0.1154 | 37.21% |
| Nonlinear Diffusion | 0.2420 | 0.1498 | 38.10% |

Table 60: *Sample density=80, train data size=20*

| Dataset | GKN | Ours | Error Reduction |
|---|---|---|---|
| Darcy Flow | 0.4937 | 0.3709 | 24.87% |
| Advection | 0.7770 | 0.5318 | 31.56% |
| Eikonal | 0.1835 | 0.1170 | 36.24% |
| Nonlinear Diffusion | 0.2387 | 0.1401 | 41.31% |

Table 61: *Sample density=90, train data size=20*

| Dataset | GKN | Ours | Error Reduction |
|---|---|---|---|
| Darcy Flow | 0.4920 | 0.3638 | 26.06% |
| Advection | 0.8157 | 0.5334 | 34.61% |
| Eikonal | 0.1847 | 0.1140 | 38.28% |
| Nonlinear Diffusion | 0.2313 | 0.1329 | 42.54% |

## B.6 Error reductions for training data size=10

Table 62: *Sample density=100, train data size=10*

| Dataset | GKN | Ours | Error Reduction |
|---|---|---|---|
| Darcy Flow | 0.5692 | 0.4673 | 17.90% |
| Advection | 1.0858 | 0.6738 | 37.94% |
| Eikonal | 0.2061 | 0.1402 | 31.98% |
| Nonlinear Diffusion | 0.2531 | 0.1691 | 33.19% |

Table 63: *Sample density=200, train data size=10*

| Dataset | GKN | Ours | Error Reduction |
|---|---|---|---|
| Darcy Flow | 0.4593 | 0.3431 | 25.30% |
| Advection | 1.0516 | 0.6407 | 39.07% |
| Eikonal | 0.2018 | 0.1276 | 36.77% |
| Nonlinear Diffusion | 0.2500 | 0.1605 | 35.80% |

Table 64: *Sample density=300, train data size=10*

| Dataset | GKN | Ours | Error Reduction |
|---|---|---|---|
| Darcy Flow | 0.4014 | 0.3233 | 19.46% |
| Advection | 0.9618 | 0.6674 | 30.61% |
| Eikonal | 0.1955 | 0.1246 | 36.27% |
| Nonlinear Diffusion | 0.2548 | 0.1498 | 41.21% |

Table 65: *Sample density=400, train data size=10*

| Dataset | GKN | Ours | Error Reduction |
|---|---|---|---|
| Darcy Flow | 0.3750 | 0.3109 | 17.09% |
| Advection | 1.0257 | 0.6733 | 34.36% |
| Eikonal | 0.1913 | 0.1203 | 37.11% |
| Nonlinear Diffusion | 0.2430 | 0.1411 | 41.93% |

Table 66: *Sample density=500, train data size=10*

| Dataset | GKN | Ours | Error Reduction |
|---|---|---|---|
| Darcy Flow | 0.3764 | 0.2842 | 24.50% |
| Advection | 0.9785 | 0.7011 | 28.35% |
| Eikonal | 0.1905 | 0.1155 | 39.37% |
| Nonlinear Diffusion | 0.2383 | 0.1464 | 38.56% |

Table 67: *Sample density=600, train data size=10*

| Dataset | GKN | Ours | Error Reduction |
|---|---|---|---|
| Darcy Flow | 0.3645 | 0.2886 | 20.82% |
| Advection | 1.0162 | 0.7271 | 28.45% |
| Eikonal | 0.1929 | 0.1255 | 34.94% |
| Nonlinear Diffusion | 0.2360 | 0.1440 | 38.98% |

Table 68: *Sample density=700, train data size=10*

| Dataset | GKN | Ours | Error Reduction |
|---|---|---|---|
| Darcy Flow | 0.3522 | 0.2768 | 21.41% |
| Advection | 0.9918 | 0.6590 | 33.56% |
| Eikonal | 0.1929 | 0.1180 | 38.83% |
| Nonlinear Diffusion | 0.2289 | 0.1363 | 40.45% |

Table 69: *Sample density=800, train data size=10*

| Dataset | GKN | Ours | Error Reduction |
|---|---|---|---|
| Darcy Flow | 0.3400 | 0.2781 | 18.21% |
| Advection | 1.0006 | 0.7109 | 28.95% |
| Eikonal | 0.1909 | 0.1210 | 36.62% |
| Nonlinear Diffusion | 0.2304 | 0.1406 | 38.98% |

Table 70: *Sample density=900, train data size=10*

| Dataset | GKN | Ours | Error Reduction |
|---|---|---|---|
| Darcy Flow | 0.3356 | 0.2812 | 16.21% |
| Advection | 1.0374 | 0.6862 | 33.85% |
| Eikonal | 0.1913 | 0.1245 | 34.92% |
| Nonlinear Diffusion | 0.2319 | 0.1420 | 38.77% |

Table 71: *Sample density=1000, train data size=10*

| Dataset | GKN | Ours | Error Reduction |
|---|---|---|---|
| Darcy Flow | 0.3345 | 0.2787 | 16.68% |
| Advection | 1.0026 | 0.6668 | 33.49% |
| Eikonal | 0.1956 | 0.1286 | 34.25% |
| Nonlinear Diffusion | 0.2346 | 0.1204 | 48.68% |

Table 72: *Sample density=20, train data size=10*

| Dataset | GKN | Ours | Error Reduction |
|---|---|---|---|
| Darcy Flow | 0.6578 | 0.5963 | 9.35% |
| Advection | 1.1149 | 0.7329 | 34.26% |
| Eikonal | 0.2144 | 0.1797 | 16.18% |
| Nonlinear Diffusion | 0.3280 | 0.3074 | 6.28% |

Table 73: *Sample density=30, train data size=10*

| Dataset | GKN | Ours | Error Reduction |
|---|---|---|---|
| Darcy Flow | 0.6580 | 0.5529 | 15.97% |
| Advection | 1.1036 | 0.6746 | 38.87% |
| Eikonal | 0.2073 | 0.1606 | 22.53% |
| Nonlinear Diffusion | 0.3017 | 0.2735 | 9.35% |

Table 74: *Sample density=40, train data size=10*

| Dataset | GKN | Ours | Error Reduction |
|---|---|---|---|
| Darcy Flow | 0.6207 | 0.5169 | 16.72% |
| Advection | 1.1147 | 0.6941 | 37.73% |
| Eikonal | 0.2090 | 0.1521 | 27.22% |
| Nonlinear Diffusion | 0.2920 | 0.2439 | 16.47% |

Table 75: *Sample density=50, train data size=10*

| Dataset | GKN | Ours | Error Reduction |
|---|---|---|---|
| Darcy Flow | 0.5909 | 0.4944 | 16.33% |
| Advection | 1.0714 | 0.6219 | 41.95% |
| Eikonal | 0.2063 | 0.1498 | 27.39% |
| Nonlinear Diffusion | 0.2788 | 0.2048 | 26.54% |

Table 76: *Sample density=60, train data size=10*

| Dataset | GKN | Ours | Error Reduction |
|---|---|---|---|
| Darcy Flow | 0.5624 | 0.4762 | 15.33% |
| Advection | 1.1333 | 0.6342 | 44.04% |
| Eikonal | 0.2051 | 0.1483 | 27.69% |
| Nonlinear Diffusion | 0.2751 | 0.1963 | 28.64% |

Table 77: *Sample density=70, train data size=10*

| Dataset | GKN | Ours | Error Reduction |
|---|---|---|---|
| Darcy Flow | 0.5629 | 0.4682 | 16.82% |
| Advection | 1.1024 | 0.6654 | 39.64% |
| Eikonal | 0.2039 | 0.1450 | 28.89% |
| Nonlinear Diffusion | 0.2609 | 0.1776 | 31.93% |

Table 78: *Sample density=80, train data size=10*

| Dataset | GKN | Ours | Error Reduction |
|---|---|---|---|
| Darcy Flow | 0.5795 | 0.4561 | 21.29% |
| Advection | 1.0573 | 0.6365 | 39.80% |
| Eikonal | 0.2041 | 0.1412 | 30.82% |
| Nonlinear Diffusion | 0.2600 | 0.1777 | 31.65% |

Table 79: *Sample density=90, train data size=10*

| Dataset | GKN | Ours | Error Reduction |
|---|---|---|---|
| Darcy Flow | 0.5824 | 0.4362 | 25.10% |
| Advection | 1.1247 | 0.6612 | 41.21% |
| Eikonal | 0.2047 | 0.1401 | 31.56% |
| Nonlinear Diffusion | 0.2597 | 0.1693 | 34.81% |

## C  Datasets Descriptions

### C.1  Darcy Flow

We considered a steady-state 2D Darcy Flow equation (Li et al., 2020a),

$$-\nabla \cdot (a(x)\nabla u(x)) = f(x) \quad x \in (0,1)^2,$$
$$u(x) = 0 \quad x \in \partial(0,1)^2, \tag{23}$$

where $u(\mathbf{x})$ is the velocity of the flow, $a(\mathbf{x})$ characterizes the conductivity of the media, and $f(\mathbf{x})$ is the source function that can represent flow sources or sinks within the domain. In the experiment, our goal is to predict the solution $u$ given the external source $f$. To this end, we fixed the conductivity $a$, which is generated by first sampling a Gauss random field $\alpha$ in the domain and then applying a thresholding rule: $a(\mathbf{x}) = 4$ if $\alpha(\mathbf{x}) < 0$, otherwise $a(\mathbf{x}) = 12$. We then used another Gauss random field to generate samples of $f$. We followed (Li et al., 2020a) to solve the PDE using a second-order finite difference solver and collected the source and solution at a $128 \times 128$ grid.

### C.2  Nonlinear Diffusion PDE

We next considered a nonlinear diffusion PDE,

$$\partial_t u(x,t) = 10^{-2}\partial_{xx}u(x,t) + 10^{-2}u^2(x,t) + f(x,t),$$
$$u(-1,t) = u(1,t) = 0, \quad u(x,0) = 0, \tag{24}$$

where $(x,t) \in [-1,1] \times [0,1]$. Our objective is to predict the solution function $u$ given the source function $f$. We used the solver provided in (Lu et al., 2022), and discretized both the input and output functions at a $128 \times 128$ grid. The source $f$ was sampled from a Gaussian process with an isotropic square exponential (SE) kernel for which the length scale was set to 0.2.

### C.3  Eikonal Equation

Third, we employed the Eikonal equation, widely used in geometric optics and wave modeling. It describes given a wave source, the propagation of wavefront across the given media where the wave speed can vary at different locations. The equation is as follows,

$$|\nabla u(\mathbf{x})| = \frac{1}{f(\mathbf{x})}, \quad \mathbf{x} \in [0,256] \times [0,256], \tag{25}$$

where $u(\mathbf{x})$ is the travel time of the wavefront from the source to location $\mathbf{x}$, $|\cdot|$ denotes the Euclidean norm, and $f(\mathbf{x}) > 0$ is the speed of the wave at $\mathbf{x}$.

In the experiment, we set the wave source at $(0,10)$. The goal is to predict the travel time $u$ given the heterogeneous wave speed $f$. We sampled an instance of $f$ using the expression:

$$f(\mathbf{x}) = \max(g(\mathbf{x}), 0) + 1.0,$$

where $g(\cdot)$ is sampled from a Gaussian process using the isotropic SE kernel with length-scale 0.1. We employed the `eikonalfm` library (https://github.com/kevinganster/eikonalfm/tree/master) that implements the Fast Marching method Sethian (1999) to compute the solution $u$.

### C.4  Poisson Equation

Fourth, we considered a 2D Poisson Equation,

$$-\Delta u = f, \quad \text{in } \Omega = [0,1]^2, \quad u|_{\partial D} = 0. \tag{26}$$

where $\Delta$ is the Laplace operator. The solution is designed to take the form, $u(x_1, x_2) = \frac{1}{\pi K^2} \sum_{i=1}^{K} \sum_{j=1}^{K} a_{ij}(i^2 + j^2)^r \sin(i\pi x_1)\cos(j\pi x_2)$, and $f(x_1, x_2)$ is correspondingly computed via the equation. To generate the dataset, we set $K = 5$ and $r = 0.5$, and independently sampled each element $a_{ij}$ from a uniform distribution on $[0,1]$.

### C.5 Advection Equation

Fifth, we considered a wave advection equation,

$$\frac{\partial u}{\partial t} + \frac{\partial u}{\partial x} = f, \quad x \in [0,1], \quad t \in [0,1]. \tag{27}$$

The solution is represented by a kernel regressor, $u(\mathbf{x}) = \sum_{j=1}^{M} w_j k(\mathbf{x}, \mathbf{z}_j)$, and the source $f$ is computed via the equation. To collect instances of $(f, u)$, we used the square exponential (SE) kernel with length-scale 0.25. We randomly sampled the locations $\mathbf{z}_j$ from the domain and the weights $w_j$ from a standard normal distribution.

