# OpenReview forum: "Graph-Based Operator Learning from Limited Data on Irregular Domains"
_TMLR — Rejected by TMLR_

### Review · Reviewer_A5XR · 2026-02-03

**Summary Of Contributions:**

**Summary**

This paper introduces GOLA (Graph-based Operator Learning with Attention), a neural operator framework designed to learn PDE solution operators on irregularly sampled spatial domains and under limited data regimes. The method represents spatial samples as nodes in a proximity graph, applies a learnable Fourier-based encoder to embed input functions into a spectral representation, and then uses attention-enhanced graph neural networks to propagate both local and global information before predicting the output solution.
The key technical contributions are:
* A learnable Fourier encoder that maps irregularly sampled input functions into a frequency space using trainable complex-valued bases, avoiding the fixed-grid requirement of FFT-based neural operators.
* A hybrid GNN architecture that combines message passing with multi-head self-attention to capture both short-range geometric interactions and long-range dependencies on irregular graphs.
* An empirical evaluation across four standard 2D PDE benchmarks (Darcy Flow, Advection, Eikonal, and Nonlinear Diffusion), demonstrating improved accuracy over DeepONet, AFNO, and Graph Kernel Networks, particularly in low-data and sparse-sampling regimes.
* A resolution and sampling-density generalization study showing robustness to changes in both training and test discretizations.

**Strengths**

* S1: The paper tackles a well-motivated and practically important limitation of existing neural operators, namely their reliance on regular grid discretizations, by enabling learning on irregularly sampled domains.

* S2: It proposes a coherent integration of spectral representations with graph-based and attention-based learning, resulting in a flexible architecture that is well-suited to non-uniform spatial data.

* S3: The experimental evaluation is extensive across multiple PDEs, training set sizes, and sampling densities, and provides clear empirical evidence that the proposed method is more data-efficient than strong baseline models.

**Weaknesses**

* W1: The theoretical analysis mainly reiterates existing universal approximation arguments for neural operators and GNNs, offering limited insight that is specific to the proposed GOLA architecture.

* W2: The model is architecturally complex, yet ablation studies are limited, making it difficult to disentangle the individual contributions of the Fourier encoder, attention mechanisms, and message aggregation design choices.

* W3: Experimental comparisons are restricted to a narrow set of baselines, and the paper does not fully evaluate against more recent graph-based or attention-driven neural operator models, while uneven writing quality and occasional notation ambiguities further reduce clarity.

**Audience:**

Yes

**Audience Explanation:**

The paper addresses operator learning on irregular domains, a problem of clear interest to researchers in scientific machine learning, neural operators, and PDE surrogate modeling. Its graph-based formulation and data-efficient learning results are likely to be relevant to both method developers and practitioners working with non-uniform or unstructured simulation data.

**Claims And Evidence:**

Yes

**Claims Explanation:**

The submission’s main claims are supported by clear and convincing empirical evidence. The paper evaluates the proposed method on multiple PDE benchmarks and systematically studies performance across different training set sizes, sampling densities, and resolutions. Consistent improvements over established baselines such as DeepONet, AFNO, and GKN are reported through quantitative results and visualizations. Although additional ablations and broader baseline comparisons would further strengthen the evidence, the existing experimental results are sufficient to substantiate the core claims of the paper.

**Requested Changes:**

1. Add targeted ablation studies that isolate the contributions of the learnable Fourier encoder, attention mechanism, and message aggregation strategy to clarify which components are essential for the observed gains.

2. Expand the set of baselines to include more recent graph-based and attention-based neural operator methods to better contextualize performance improvements [1,2].

3. Improve the theoretical section by adding analysis or discussion that provides insights specific to the proposed architecture, beyond standard universal approximation arguments.

4. Strengthen the presentation by clarifying notation, correcting grammatical issues, and streamlining the methodology description for improved readability.

5. Include additional experiments or discussion on more challenging or realistic irregular domains, such as unstructured meshes or higher-dimensional problems, to better demonstrate general applicability.

---
[1] Li, Z., Song, H., Xiao, D., Lai, Z., & Wang, W. (2025). Harnessing scale and physics: A multi-graph neural operator framework for pdes on arbitrary geometries.

[2] Sarkar, S., & Chakraborty, S. (2025). Spatio-spectral graph neural operator for solving computational mechanics problems on irregular domain and unstructured grid.

---

> ### Author Response · Authors · 2026-03-04
>
> We thank the reviewer for the constructive comments.
> > Improve the theoretical section by adding analysis or discussion that provides insights specific to the proposed architecture, beyond standard universal approximation arguments.
>
> The Fourier encoder introduces a global functional representation prior to graph-based processing. Classical neural operators such as FNO exploit spectral representations to efficiently capture global structure in solution fields. In the proposed architecture, the learnable Fourier basis projects irregularly sampled input functions into a spectral feature space before graph message passing.
> This spectral projection allows the model to represent long-range functional dependencies using a small number of basis functions, which can improve approximation efficiency under sparse sampling. In contrast, purely local graph neural networks must propagate information across many message passing layers to capture global structure.
>
> The architecture can be interpreted as a decomposition of operator learning into three stages:
> (1) global spectral encoding through the Fourier basis,
> (2) local geometric reasoning through graph message passing, and
> (3) global interaction refinement through attention-based aggregation.
> This combination allows the model to capture both local relational structures and long-range dependencies across irregular spatial samples. In particular, the attention module enables nodes to exchange information beyond their immediate graph neighborhoods, mitigating the limited receptive field typically associated with message passing neural networks.
>
> > Strengthen the presentation by clarifying notation, correcting grammatical issues, and streamlining the methodology description for improved readability.
>
> Response: I have added a table of Summary of Notations in Appendix in the paper. I also have strengthed mathematical rigorousness in the paper.

---

### Review · Reviewer_f5yN · 2026-02-07

**Summary Of Contributions:**

This paper proposes GOLA (Graph-based Operator Learning with Attention), a neural operator framework designed to learn solution operators of PDEs from irregularly sampled spatial domains and limited data. The work addresses a key limitation of popular operator learning methods such as DeepONet and Fourier Neural Operator (FNO), which typically assume structured grid discretizations and struggle under sparse or non-uniform sampling.

The main contributions of the paper are:

$\textbf{Graph-based operator learning on irregular domains}$:
The authors represent spatial samples as nodes in a proximity graph, enabling operator learning on arbitrary geometries and non-uniform point clouds. This formulation naturally generalizes beyond regular Cartesian grids and is well suited for real-world scientific settings.

$\textbf{Learnable Fourier encoder for irregular sampling}$
A central novelty is a Fourier-based encoder with learnable complex frequencies, which projects irregularly sampled input functions into a spectral representation before graph processing. This allows the model to capture global structure while remaining mesh-free.

$\textbf{Attention-enhanced message passing}$
The framework integrates multi-head self-attention with graph message passing to model both local interactions and long-range dependencies, improving expressivity over standard GNN-based operator learners.

$\textbf{Theoretical justification via operator approximation}$
The paper provides a universal approximation argument, showing that GOLA can approximate continuous nonlinear operators between Banach spaces under sufficient capacity, leveraging results from neural operator theory and GNN expressivity.

$\textbf{Extensive empirical validation under data scarcity}$
GOLA is evaluated on four 2D PDE benchmarks (Darcy Flow, Advection, Eikonal, Nonlinear Diffusion) across a wide range of training sizes and sampling densities. The method consistently outperforms DeepONet, AFNO, and Graph Kernel Networks (GKN), particularly in low-data and sparse-sampling regimes.

Strengths include strong motivation, consistent empirical gains under challenging regimes, and a principled hybrid of spectral and graph-based learning. Weaknesses include limited ablation of architectural components and a largely high-level theoretical analysis without operator-specific rates.

**Audience:**

Yes

**Audience Explanation:**

The paper is highly relevant to the TMLR audience, particularly researchers working on: i) Neural operators and operator learning theory,
ii) Scientific machine learning and surrogate modeling, iii) Graph neural networks for PDEs and physical systems, iv) Learning under sparse, irregular, or non-Euclidean data.
The combination of graph representations, spectral encoding, and attention mechanisms places this work at the intersection of several active research directions. Moreover, the strong focus on data-limited regimes aligns well with practical constraints in scientific and engineering applications, making the findings broadly interesting and impactful.

**Broader Impact Concerns:**

None. The work focuses on surrogate modeling for physical systems and does not raise specific ethical concerns.

**Claims And Evidence:**

Yes

**Claims Explanation:**

The methodological design directly addresses the stated limitation of grid-based operators by operating on graphs constructed from irregular samples, which is clearly illustrated in the architecture and graph visualizations. The Fourier encoder is rigorously defined using learnable complex bases, and its role in improving expressivity under sparse sampling is empirically validated. The experimental evaluation is extensive, covering: a) Multiple PDE families with distinct characteristics, b) A wide range of training data sizes (as few as 5–10 samples),
c) Varying sampling densities and resolution generalization d) Across all settings, GOLA consistently achieves lower relative L2 error than strong baselines, with especially large margins in data-scarce regimes.

The theoretical analysis, while high-level, correctly situates GOLA within the universal approximation framework for neural operators and GNNs, lending conceptual support to the architecture. Overall, the empirical trends are consistent, reproducible across seeds, and aligned with the paper’s claims regarding robustness, data efficiency, and irregular-domain generalization.

**Requested Changes:**

$\textbf{Clarify novelty relative to prior graph-based neural operators}$: While the paper combines GNNs, attention, and Fourier features, the distinction from prior works such as GKN, PDE-GCN, and attention-based operator learners should be sharpened. A more explicit architectural and conceptual comparison would strengthen the contribution.

$\textbf{Ablation studies on key components}$: The paper would benefit from controlled ablations isolating:
i) the learnable Fourier encoder vs. fixed Fourier features, ii) attention vs. pure message passing, iii) graph connectivity radius sensitivity.
This would clarify which components drive the observed performance gains.

$\textbf{More precise theoretical positioning}$:
The approximation result largely restates known universal approximation arguments. Clarifying whether GOLA offers any advantages in approximation rates, stability, or sample efficiency would improve the theoretical contribution.

$\textbf{Few more general points}$: i) Include runtime and memory comparisons against baselines to complement accuracy results. ii) Discuss extension to time-dependent or higher-dimensional PDEs more explicitly.

---

> ### Author Response · Authors · 2026-03-04
>
> We thank the reviewer for the constructive comments.
> > While the paper combines GNNs, attention, and Fourier features, the distinction from prior works such as GKN, PDE-GCN, and attention-based operator learners should be sharpened. A more explicit architectural and conceptual comparison would strengthen the contribution.
>
> Response: We have the idea about fourier encoder before graph learning, attention inside graph message passing, and combination of spectral,graph,attention.
>
> Specifically, the proposed GOLA architecture integrates three complementary components:
> (1) a learnable Fourier encoder that injects global spectral structure into the representation,
> (2) graph-based message passing that captures local geometric relationships on irregular domains, and
> (3) multi-head self-attention and attention-weighted aggregation that enable global dependency modeling across nodes.
> This unified design allows the model to simultaneously exploit spectral priors, relational inductive biases, and long-range interactions while remaining applicable to irregularly sampled spatial domains.
>
> I have updated it in the paper.
>
>
>
> > The approximation result largely restates known universal approximation arguments. Clarifying whether GOLA offers any advantages in approximation rates, stability, or sample efficiency would improve the theoretical contribution.
>
> The result primarily establishes the expressivity of the proposed architecture by showing that the GOLA model can approximate continuous operators between Banach spaces under sufficient capacity. Similar universal approximation results have been established for other neural operator architectures such as DeepONet.
>
> The contribution of GOLA lies not in altering the theoretical approximation class, but in providing a more effective inductive bias for learning operators defined on irregular spatial domains. By combining spectral representations, graph-based message passing, and attention-based global aggregation, the architecture improves the ability to capture both local geometric structure and long-range functional dependencies. Empirically, this design leads to improved sample efficiency and generalization under sparse and irregular sampling regimes.

---

### Review · Reviewer_ZxUk · 2026-02-18

**Summary Of Contributions:**

The article introduces a graph-based operator learning technique which is in particular suitable for non-grid training data. The architecture is based on a Fourier-based encoder, Graph neural networks and attention mechanism. In this way, the proposed method shows superior performance in learning the solution for several 2D PDE system in contrast to traditional operator learning approaches.

Strengths:
- Having non-grid input data is a common situation in real-world engineering setting due to technical limitations. The article tries to address this limitation, which can have a substantial impact.

Weaknesses:
- The model architecture is introduced in a step-by-step manner, however there is little to no explanation WHY the authors choose the particular structure. It would be beneficial for the reader to provide more context and explanation here.
- Several details are not clearly communicated, and variables are often not introduced and defined. Mathematical rigorousness would strengthen the article.
- It is unclear if the proposed architecture is explicitly beneficial for non-grid data or improves the performance in general. Unfortunately, the authors one little investigate in these important details. Furthermore, the role of some design variable is not properly evaluated.

**Audience:**

Yes

**Audience Explanation:**

The paper might address the issue of how the model performance can be kept high with non-grid data. This is a relevant problem since in practice measurements can be off grid. In this way, the proposed method tries to close a relevant and interesting research gap.

**Claims And Evidence:**

No

**Claims Explanation:**

The article claims that “traditional” methods have issues with non-grid data, which, however, is not fully supported by evidence. The simulation needs more detailed analysis (as stated in the requested changes) to be convincing.

**Requested Changes:**

Major:
- The simulation section does not evaluate if the increase of performance is due to the advanced capability of handling non-grid data or the architecture itself. Could the authors compare the performance of GOLA vs the baselines on one example with regular grid data. If the performance is similar, I would like to see the performance of all methods using non-grid data, perform some kind of function approximation suitable for small data sets (e.g. a Gaussian process) to interpolate on a regular grid. These points should then be used as input data. Also visualize how the solution of the PDEs look like.

- The impact of r is rarely analyzed. Please perform some ablation studies how the performance is impacted by r.

Minor:
- Add 1-2 sentences to each paragraph in the related work section naming the research gap
- 3.1 Mention that f is element of F (Banach space) and the same for u (I think that this is indicated here but not said)
- 3.1 “While existing approaches such as DeepONet and FNO have demonstrated strong performance, they typically rely on structured, grid-based discretizations of the domain” if they “typically” rely, is there a version which does not rely on grid-based points? If so, please discuss. If not, delete.
- 3.2 “To represent PDE solutions over irregular domains, we begin by randomly sampling a subset of points […]” This seems more like an example instead of the general case. In general, we just have a set of points (maybe from measurements), somehow distributed in the spatial domain. Please keep it general here and move this specific approach to the simulation section (in case you use it).
- 3.3: “Given the input f\in R^[…]” f should live in a function space as defined in 3.1. Please clarify.
- 3.3 Please clarify the meaning of C_in and C_out.
- (8) Doublecheck, there seems to be a missing ||. Also discuss where the equation in coming from (your contribution or add reference)
- (12) What is \mathcal{K}_h? There are several undefined variables in the manuscript. Please introduced and define.
- Section 3: Add more explanation WHY this architecture has been chosen. Does it built on existing approaches? Is this architecture entirely new?
- Section 4: “Since \Omega is compact, by increasing N the point cloud {x_i} becomes dense. Please be more specific as I can sample infinite many points in a subset of the spatial domain such that it’s not going to be dense (e.g. sample all points at the same location)
- Section 4: Clarify the roll of \mathcal{F}_\sigma. What happens if f is in \mathcal{F} but not \mathcal{F}_\sigma? Furthermore, since the authors use continuity in the argumentation, what happens if the solution u is discontinues (there seems to be no restriction in the problem statement)
- Section 5: “From Table 2, we use 100 training data” -> training points

---

> ### Author Response · Authors · 2026-03-04
>
> We thank the reviewer for the constructive comments.
> >Question: The model architecture is introduced in a step-by-step manner, however there is little to no explanation WHY the authors choose the particular structure. It would be beneficial for the reader to provide more context and explanation here.
>
> Response: I have added the following in the paper.
> Existing neural operators such as FNO rely on structured grids, which limits their applicability to irregular domains.
> The reasons why we constrcut graph are as follows:
> (1) Irregular spatial samples naturally form a point cloud;
> (2) Graphs provide permutation invariance and geometric flexibility;
> (3) Message passing enables local relational inductive bias;
> (4) Graphs allow resolution-agnostic generalization.
> Thus, the graph formulation directly addresses the limitation of grid-dependent neural operators.
>
> Classical FNO performs global convolution in Fourier space but requires uniform grids for FFT.
> The motivations why we design Fourier Encoder are that
> (1) Fourier bases provide a global functional prior;
> (2) Spectral representations are resolution-independent;
> (3) Learnable frequencies allow adaptation to irregular sampling;
> (4) Fourier Encoder injects global structure before local message passing.
> Without this spectral encoder, purely local GNNs struggle to capture long-range dependencies efficiently, especially in low-data regimes. Thus, the Fourier Encoder introduces global inductive bias while remaining mesh-free.
>
> Standard GNNs are limited by local receptive fields, over-smoothing with depth, and difficulty modeling global interactions.
> Message passing mechanisms are particularly effective for modeling strong local geometric structures.
> Multi-head self-attention enables the modeling of global dependencies by allowing each node to attend to all others.
> Attention-weighted aggregation introduces adaptivity by assigning different importance weights to neighboring nodes during feature aggregation.
> Under sparse sampling and limited training data, purely local models struggle to extrapolate global structure. The spectral encoder supplies a compact global representation, while attention enhances expressivity without requiring deep stacking. Empirically, this design leads to improved generalization and lower error in data-scarce regimes.
> > Question: Several details are not clearly communicated, and variables are often not introduced and defined. Mathematical rigorousness would strengthen the article.
>
> Response: I have added a table of Summary of Notations in Appendix in the paper. I also have strengthed mathematical rigorousness in the paper.
> > Question: 3.3: “Given the input f\in R^[…]” f should live in a function space as defined in 3.1. Please clarify.
>
> Response: Thank you for pointing this out. In Section 3.1 we define $f$ as an element of the function space $\mathcal{F}$. In Section 3.3 the model operates on discrete observations of this function. We revised the text to clarify that the tensor $f \in \mathbb{R}^{B \times C_{in} \times N}$ corresponds to sampled function values $f(x_i)$ at spatial points $\{x_i\}_{i=1}^{N}$, rather than the function itself.
> > 3.3 Please clarify the meaning of C_in and C_out.
>
> Response: $C_{in}$ is the number of input feature channels per spatial point, $C_{out}$ is the number of output feature channels produced by the Fourier encoder
> > (12) What is \mathcal{K}_h? There are several undefined variables in the manuscript. Please introduced and define.
>
> Response: I have updated it in the paper. I have added a table of Summary of Notations in Appendix in the paper.
> > Section 4: “Since $\Omega$ is compact, by increasing N the point cloud ${x_i}$ becomes dense. Please be more specific as I can sample infinite many points in a subset of the spatial domain such that it’s not going to be dense (e.g. sample all points at the same location)
>
> Response: I have updated it in the paper.
>
> > Section 4: Clarify the roll of \mathcal{F}\sigma. What happens if f is in \mathcal{F} but not \mathcal{F}\sigma? Furthermore, since the authors use continuity in the argumentation, what happens if the solution u is discontinues (there seems to be no restriction in the problem statement)
>
> Response: $\mathcal{F}_\delta$ is a compact subset of the function space.
>
> The approximation guarantee holds for functions within the compact subset $\mathcal{F}_\delta$. Functions outside this set are not covered by the theoretical guarantee.
>
> We assume the solution operator maps into $\mathcal{U} \subset L^2(\Omega)$ which allows the framework to accommodate functions that may exhibit discontinuities while remaining square-integrable.
>
> I have updated them in the paper.
>
>
> > Section 5: “From Table 2, we use 100 training data” -> training points
>
> Response: The training data size is 100. Then we sample 20, 500, 1000 data points.

---

### Decision · Action_Editor_7MzK · 2026-04-06

**Recommendation:** Reject

**Additional Comments:**

The authors should consider a more careful incorporation of the reviewers' feedback regarding clarity and notation during the revision. While the table in the appendix is a useful first step, handling the notation throughout the paper would improve readability.

**Audience:**

Yes

**Audience Explanation:**

Operator learning is an important topic in scientific machine learning. Irregularly spaced grid are a common setting, and this paper proposes a method for this setting with encouraging results.

**Claims And Evidence:**

No

**Claims Explanation:**

The authors introduce GOLA, an architecture for operator learning on irregularly spaced grids. The architecture consists of a Fourier encoder, message passing, and a transformer. The authors compare this architecture against three existing baselines.

There are several shortcomings that prevent this paper from meeting the "accurate, convincing, and clear evidence" criterion of TMLR:

1. The performance of GOLA conflates gains from architectural capacity and non-grid capabilities. While the reviewers suggested an experiment towards disentangling these effects, the authors did not provide an experiment in the rebuttal.

2. Minimal ablation. GOLA contains many complex components which, though justified through intuitions in the methods section, are not thoroughly justified through ablations and sensitivity studies in the experiments section. All reviewers wished to see more ablations; however the authors did not follow through during the rebuttal.

3. Comparison against more modern baselines. Li et al. (2025) and Sarkar & Chakraborty (2025) are relevant baselines that the authors should consider comparing against to contextualize their results.

**Resubmission Of Major Revision:**

The authors may consider submitting a major revision at a later time.